# ON THE EXPRESSIVENESS AND LEARNING OF RELATIONAL NEURAL NETWORKS ON HYPERGRAPHS

## ABSTRACT

This paper presents a framework for analyzing the expressiveness and learning of relational models applied to hypergraph reasoning tasks. We start with a general framework that unifies several relational neural network architectures: graph neural networks, neural logical machines, and transformers. Our first contribution is a fine-grained analysis of the expressiveness of these neural networks, that is, the set of functions that they can realize and the set of problems that they can solve. Our result is a hierarchy of problems they can solve, defined in terms of various hyperparameters such as depth and width. Next, we analyze the learning properties of these neural networks, especially focusing on how they can be trained on a small graphs and generalize to larger graphs. Our theoretical results are further supported by the empirical results illustrating the optimization and generalization of these models based on gradient-descent training.

## 1 INTRODUCTION

Reasoning over graph-structured data is an important task in various applications, including molecule analysis, social network modeling, and knowledge graph reasoning (Gilmer et al., 2017; Schlichtkrull et al., 2018; Liu et al., 2017). While we have seen great success of various relational neural networks, such as Graph Neural Networks (GNNs; Scarselli et al., 2008), Neural Logical Machines (NLM; Dong et al., 2019), and Transformers (Vaswani et al., 2017) in a variety of applications (Battaglia et al., 2018; Merkwirth & Lengauer, 2005; Veličković et al., 2020), we do not yet have a full understanding of how different design parameters, such as the depth of the neural network, affects the expressiveness of these models, or how effectively these models generalize from data.

In this paper, we develop a general framework, *generalized relational neural networks* (RelNNs), that unifies a large class of relational neural networks including GNNs, NLMs, and Transformers. Unlike previous work that analyzes the expressiveness of graph neural networks (Xu et al., 2019; 2020; Barceló et al., 2020), our framework applies to hypergraphs, i.e., graphs that have representations that directly relate groups of more than two nodes.

Next, we quantify the *expressiveness* of generalized relational neural networks in terms of their structural parameters; in particular, the maximum arity of hyperedges and the depth of the network. We formally proved the "if and only if" conditions for the expressive power w.r.t. the arity. That is, $k$-ary hyper-graph relational neural networks are sufficient and necessary for realizing FOC-$k$, a fragment of first-order logic which involves at most $k$ variables. This is a helpful result because now we can determine whether a RelNN can solve a problem by thinking whether a logic formula can represent the solution to this problem. Next, we formally described the relationship between expressiveness and non-constant-depth RelNNs. We proposed a conjecture about the "depth hierarchy," and we connect the potential proof of this conjecture to distributed computing literature. We also develope an upper bound for RelNNs. This is a very interesting result: applying a $k$-ary hyper-graph RelNN for more than $O(n^{k-1})$ iterations has no effect on the expressiveness.

Finally, we study the learning capabilities of RelNNs. We prove, under certain realistic assumptions, it is possible to train a RelNN on a finite set of graphs, and it will generalize to arbitrarily large graphs. This is an outcome due to the weight-sharing nature of RelNNs. Next, we proved a finer-grained sample-complexity bound for RelNNs. This result theoretically suggests that using max aggregation function is provably better than sum aggregation function when realizing a first-order logic formula.

We hope our work can serve as a foundation for designing relational neural networks: to solve a specific problem, what arity do you need? What depth do you need? What aggregation function should you use? Will my model have structural generalization (i.e., to larger graphs)? Our theoretical

results on learning are further supported by experiments, for empirical demonstration of the theorems and for exploring the tightness of the sample-complexity bounds and effectiveness of different aggregation functions. Specifically, we show that RelNNs with sufficient expressiveness power can be trained using gradient descent to realize the desired function. Meanwhile, when the training data distribution is uniformly distributed, RelNNs do demonstrate strong generalization to larger graphs than those seen during training. Finally, we also empirically show the generalization of RelNNs with different aggregation functions.

## 2 GRAPH REASONING AND RELATIONAL NEURAL NETWORKS

In this section, we define a very general class of relational networks and show that it can represent several important classes of relational neural networks, including graph neural networks, neural logic machines, and transformers.

### 2.1 GRAPH REASONING PROBLEM

A graph reasoning problem is a prediction task on a hypergraph $G$. It can predict a discrete or real-valued property of the whole graph, of individual nodes, or of tuples of nodes.

Much of the work on graph-structured neural-network models has focused on cases with only binary edges, but many important problems are best modeled using *hypergraphs*, in which a hyperedge can connect multiple nodes. A further generalization is to *labeled* hypergraphs, in which each hyperedge can have one of a discrete set of labels. We will make use of a still more general structure, in which hyperedges can have real and/or vector-valued labels.

A *hypergraph representation* $G$ is a tuple $(V, X)$, where $V$ is a set of entities (nodes), and $X$ is a set of *hypergraph representation functions*. Specifically, $X = \{X_0, X_1, X_2, \cdots, X_k\}$, where $X_j : (v_1, v_2, \cdots, v_j) \to \mathcal{S}$ is a function mapping every tuple of $j$ nodes to a value. We call $j$ the *arity* of the hyperedge and $k$ is the max arity of input hyperedges. The range $\mathcal{S}$ can be any set of discrete labels that describes relation type, or a scalar number (e.g., the length of an edge), or a vector. In general, we will use the arity 0 representation function $X_0(\emptyset) \to \mathcal{S}$ to represent any global properties of the graph as a whole.

A *graph reasoning function* $f$ is a mapping from a hypergraph representation $G = (V, X)$ to another hyperedge representation function $Y$ on $V$. There are several important types:
- graph classification function: computes a global label, $Y = \{Y_0\}; Y_0(\emptyset) \to \mathcal{S}' = \{0, 1\}$ for the whole graph.
- graph regression function: computes a global real output, $Y = \{Y_0\}$, where $Y_0(\emptyset) \to \mathbb{R}$.
- node classification/regression function: computes $Y = \{Y_1\}$, where $Y_1(v_0) \to \{0, 1\}$ is a binary label for each individual node.
- edge classification function: extends previous prediction notions to higher-arity hyperedges.

For example, asking whether a graph is fully connected is a graph classification problem; computing its diameter is a graph regression problem; finding the set of disconnected subgraphs of size $k$ is a $k$-ary hyperedge classification problem.

**Example (undirected graph connectivity).** The input is $G = (V, X)$, where $X = \{X_0, X_1, X_2\}$, in which $X_2$ is mapping from each pair of nodes $v_1, v_2 \in V$ to $\{0, 1\}$, indicating whether $(v_1, v_2)$ are connected. The input is an undirected graph; thus, $X_2(v_1, v_2) = X_2(v_2, v_1)$ for every $v_1, v_2$. Note that $X_0$ and $X_1$ are constant functions and not used for describing the undirected graph. The output is $Y = \{Y_0\}$, where $Y_0(\emptyset) \to \{0, 1\}$ is a binary classification label. The desired label is 1 when the graph is connected, and 0 otherwise.

### 2.2 RELATIONAL NEURAL NETWORKS

We now introduce introduce *generalized relational neural networks* (RelNNs), a general framework that unifies a set of important relational neural network models that can be trained to solve graph reasoning problems. In particular, we will describe relational neural networks following the notation and computation of neural logic machines (NLMs; Dong et al., 2019). We choose NLM as a base model because it naturally generalizes to hyperedges with arities greater than 2. As we will see later, other relational neural networks such as graph neural networks (GNNs; Scarselli et al., 2008) and transformers (Vaswani et al., 2017) are equivalent to NLMs and each other in terms of expressiveness. Showing this equivalence allows us to focus the rest of the paper on analyzing a single model type, with the understanding that the conclusions generalize to a broader class of relational neural networks.

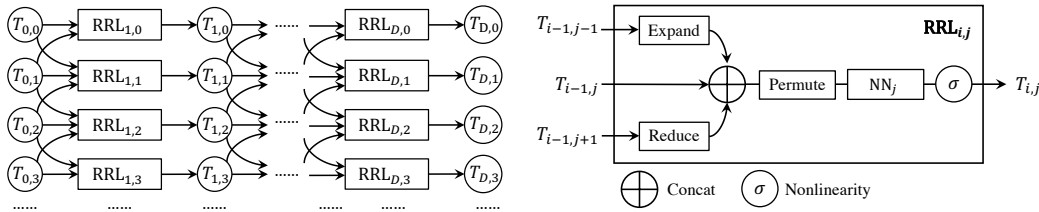

(a) The overall computation graph of a RelNN.  (b) The computation graph of a single RelNN block.

Figure 1: The overall architecture of our generalized relational neural network (RelNNs). It follows the computation graph of NLM (Dong et al., 2019) and can be applied to hypergraphs.

A RelNN is a multi-layer neural network that operates on hypergraph representations, in which the hypergraph representation functions are represented as tensors. The input is a hypergraph representation $(V, X)$. There are then several computational layers, each of which produces a hypergraph representation with nodes $V$ and a new set of representation functions. Specifically, a $B$-ary RelNN produces hypergraph representation functions with arities from 0 up to a maximum hyperedge arity of $B$. We let $T_{i,j}$ denote the tensor representation for the output at layer $i$ and arity $j$. Each entry in the tensor is a mapping from a set of node indices $(v_1, v_2, \cdots, v_j)$ to a vector in a latent space $\mathbb{R}^W$. Thus, $T_{i,j}$ is a tensor of $j + 1$ dimensions, with the first $j$ dimensions corresponding to $j$-tuple of nodes, and the last feature dimension. For convenience, we write $h_{0,.}$ for the input hypergraph representation and $h_{D,.}$ for the output of the RelNN.

Fig. 1a shows the overall architecture of RelNNs. It has $D \times B$ computation blocks, namely relational reasoning layers (RRLs). Each block $\text{RRL}_{i,j}$, illustrated in Fig. 1b, takes the output from neighboring arities in the previous layer, $T_{i-1,j-1}$, $T_{i-1,j}$ and $T_{i-1,j+1}$, and produces $T_{i,j}$. Below we show the computation of each primitive operation in an RRL.

The *expand* operation takes tensor $T_{i-1,j-1}$ (arity $j - 1$) and produces a new tensor $T_{i-1,j-1}^E$ of arity $j$. The *reduce* operation takes tensor $T_{i-1,j+1}$ (arity $j + 1$) and produces a new tensor $T_{i-1,j+1}^R$ of arity $j + 1$. Mathematically,

$$\begin{aligned}
T_{i-1,j-1}^E[v_1, v_2, \cdots, v_j] &= T_{i-1,j-1}[v_1, v_2, \cdots, v_{j-1}]; \\
T_{i-1,j+1}^R[v_1, v_2, \cdots, v_j] &= \text{Agg}_{v_{j+1}} \left\{ T_{i-1,j+1}[v_1, v_2, \cdots, v_j, v_{j+1}] \right\}.
\end{aligned}$$

Here, Agg is called the aggregation function of a RelNN. For example, a sum aggregation function takes the summation along the dimension $j + 1$ of the tensor, and a max aggregation function takes the max along that dimension.

The *concat* (concatenate) operation $\bigoplus$ is applied at the "vector representation" dimension. The *permute* operation generates a new tensor of the same arity, but it fuses the representations of hyperedges that share the same set of entities but in different order, such as $(v_1, v_2)$ and $(v_2, v_1)$. Mathematically, for tensor $X$ of arity $j$, if $Y = \text{permute}(X)$ then

$$Y[v_1, v_2, \cdots, v_j] = \underset{\sigma \in S_j}{\text{Concat}} \left\{ X[v_{\sigma_1}, v_{\sigma_2}, \cdots, v_{\sigma_j}] \right\},$$

where $\sigma \in S_j$ iterates over all permuations of $\{1, 2, \cdots j\}$. $\text{NN}_j$ is a multi-layer perceptron (MLP) applied to each entry in the tensor produced after permutation, with nonlinearity $\sigma$ (e.g., ReLU).

It is important to note that we intentionally name the MLPs $\text{NN}_j$ instead of $\text{NN}_{i,j}$. In generalized relational neural networks, for a given arity $j$, all MLPs across all layers $i$ are shared. It is straightforward to see that this "weight-shared" model can realize a "non-weight-shared" RelNN that uses different weights for MLPs at different layers when the number of layers is a constant. With a sufficiently large length of the representation vector, we can simulate the computation of applying different transformations by constructing block matrix weights. (A more formal proof is in Appendix B.1) The advantage of this weight sharing is that the network can be easily extended to a "recurrent" model. For example, we can apply the RelNN for a number of layers that is a function of $n$, where $n$ is the the number of nodes in the input graph. Thus, we will use the term *layers* and *iterations* interchangeably.

Handling high-arity features and using deeper models usually increase the computational cost. In appendix B.6, we show that the time and space complexity of RelNN $[D, B]$ is $O(Dn^B)$.

**Exanple (GNN).** The *graph neural network* (GNN) of Scarselli et al. (2008) can be described as a specific type of RelNN. First, GNNs work on graphs with maximum arity 2. In a single iteration in a

| | Classification Tasks | Regression Tasks |
|---|---|---|
| $B = 4$ | 4-Clique Detection RelNN[$O(1)$, 4] | 4-Clique Count RelNN[$O(1)$, 4] |
| $B = 3$ | Triangle Detection RelNN[$O(1)$,3] Bipartiteness RelNN[$O(\log n)$, 3]$^\star$ All-Pair Connectivity RelNN[$O(\log n)$, 3]$^\star$ All-Pair Connectivity-$k$ RelNN[$O(\log k)$, 3]$^\star$ | All-Pair Distance RelNN[$O(\log n)$, 3]$^\star$ |
| $B = 2$ | 3-Link Detection RelNN[$O(1)$, 2] 4-Link Detection RelNN[$O(1)$, 2] S-T Connectivity RelNN[$O(n)$, 2] S-T Connectivity-$k$ RelNN[$O(k)$, 2] FOC$_2$ Realization RelNN[$\cdot$, 2] (Barceló et al., 2020) | S-T Distance RelNN[$O(n)$, 2] Max Degree RelNN[$O(1)$, 2] Max Flow RelNN[$O(n^3)$, 2]$^\star$ |
| $B = 1$ | Node Color Majority: RelNN[$O(1)$, 1] | Count Red Nodes: RelNN[$O(1)$, 1] |

Table 1: The minimum depth and arity of RelNNs for solving graph classification and regression tasks. The $^\star$ symbol indicates that these are conjectured lower bounds.

GNN, we use two message-passing subroutines: (edge update) the representation associated with each edge is updated by representations of its both ends, and (node update): the representation of each node is updated by all edge presentations connected to it. This message passing scheme can easily fit in the RelNN: edge update can be realized by the *expand* and *permute* operations, while node update can be realized by the *reduce* operation.

In general, hypergraph GNNs (Morris et al., 2019) are also equivalent to RelNNs. Specifically, a GNN applied to $B$-ary hyperedges is equivalent to an NLM applied to $B + 1$-ary hyperedges. We provide proof details in Appendix B.2.

There are other approaches of realizing GNN on hypergraphs such as hypergraph convolution(Feng et al., 2019; Yadati et al., 2019; Bai et al., 2021), attention(Ding et al., 2020) and message passing(Huang & Yang, 2021). These approaches can be viewed as instances of RelNNs, but are less expressive than RelNNs with equal max arity. We provide proof details in Appendix B.3.

**Example (Transformer).** The key difference between Transformers (Vaswani et al., 2017) and GNNs is the "self-attention" opeartion, which can be viewed as a special aggregation function. In self-attention, each "sender" computes the *key* and *value*, and each "receiver" computes the *query*. Xu et al. (2019) proved such aggregation can be simulated with a *sum* aggregation function.

We will use calligraphic letters $\mathcal{M}_1, \mathcal{M}_2, \cdots$ to name RelNNs. Because even when hyperparameters such as the maximum arity and the number of iterations are fixed, a RelNN is still a *model family* $\mathcal{M}$: the weights for MLPs will be trained on some data. Furthermore, each model $M \in \mathcal{M}$ is a RelNN with a specific set of MLP weights.

## 3 EXPRESSIVENESS OF RELATIONAL NEURAL NETWORKS

A RelNN is characterized by hyperparameters $D$ (depth), and $B$ maximum arity. In this section, we will focus on $D$ and $B$, quantifying how they affect the expressiveness of a model; i.e., the set of graph reasoning functions a model family can realize. The $W$ parameter affects the precise details of what functions can be realized, as it does in a regular neural network, but does not affect the structural constraints that are our focus in this paper. When considering $D$, we will focus on the distinction between any constant depth *vs.* adaptive depth, in which $D$ depends on the number $n$ of nodes in the input graph. We will use the notation RelNN[$D$, $B$] to represent a RelNN family with depth $D$ and max arity $B$. A summary of some concrete results is in table 1 and discussed in detail in Appendix A.

Many frameworks for characterizing the expressiveness of neural networks make a distinction between models that have the capacity to use unbounded precision in intermediate representations. In all results we state below, we will indicate whether they apply to the fixed or unbounded precision case.

**Definition 3.1** (Expressiveness). We say a model family $\mathcal{M}_1$ is *at least expressive as* $\mathcal{M}_2$, written as $\mathcal{M}_1 \succeq \mathcal{M}_2$, if for all $M_2 \in \mathcal{M}_2$, there exists $M_1 \in \mathcal{M}_1$ such that $M_1$ can realize $M_2$. A model family $\mathcal{M}_1$ is *more expressive than* $\mathcal{M}_2$, written as $\mathcal{M}_1 \succ \mathcal{M}_2$, if $\mathcal{M}_1 \succeq \mathcal{M}_2$ and $\exists M_1 \in \mathcal{M}_1$, $\forall M_2 \in \mathcal{M}_2$, $M_2$ can not realize $M_1$.

## 3.1 ARITY HIERARCHY

We first aim to quantify how the maximum arity $B$ of the network's representation affects its expressiveness and find that, in short, the higher the maximum arity, the more expressive the relational neural network is. A majority of the results in this section are based on the Weisfeiler-Leman graph isomorphism test (WL-test) (Leman & Weisfeiler, 1968; Xu et al., 2019).

**Theorem 3.1** (Arity Hierarchy). For any maximum arity $B$, there exists a depth $D^*$ such that: $\forall D \geq D^*$, RelNN$[D, B + 1]$ is more expressive than RelNN$[D, B]$. This theorem applies to both fixed-precision and unbounded-precision networks.

*Proof sketch:* Our proof slightly extends the proof of Morris et al. (2019). First, the set of graphs distinguishable by RelNN$[D, B]$ is bounded by graphs distinguishable by a $D$-round order-$B$ WL-test. If models in RelNN$[D, B]$ cannot generate different outputs for two distinct hypergraphs $G_1$ and $G_2$, but there exists $M \in$ RelNN$[D, B + 1]$ that *can* generate different outputs for $G_1$ and $G_2$, then we can construct a graph classification function $f$ that RelNN$[D, B + 1]$ can realize but RelNN$[D, B]$ cannot. The full proof is described in Appendix B.4.

It is also important to quantify the minimum arity for realizing certain graph reasoning functions.

**Theorem 3.2** (FOL realization bounds). Let FOC$_B$ denote a fragment of first order logic with at most $B$ variables, extended with counting quantifiers of the form $\exists^{\geq n}\phi$, which state that there are at least $n$ nodes satisfying formula $\phi$ (Cai et al., 1992).
- (Upper Bound) Any function $f$ in FOC$_B$ can be realized by RelNN$[D, B]$ for some $D$.
- (Lower Bound) There exists a function $f \in$ FOC$_B$ such that for all $D$, $f$ cannot be realized by RelNN$[D, B - 1]$.

*Proof:* The upper bound part of the claim has been proved by Barceló et al. (2020) for $B = 2$. The results generalize easily to arbitrary $B$ because the counting quantifiers can be realized by sum aggregation. The lower bound part can be proved by applying Section 5 of Cai et al. (1992), in which they show that FOC$_B$ is equivalent to a $(B - 1)$-dimensional WL test in distinguishing non-isomorphic graphs. Given that RelNN$[D, B - 1]$ is equivalent to the $(B - 2)$-dimensional WL test of graph isomorphism, there must be an FOL$_B$ formula that distinguishes two non-isomorphic graphs that RelNN$[D, B - 1]$ cannot. Hence, FOL$_B$ cannot be realized by RelNN$[\cdot, B - 1]$.

## 3.2 DEPTH HIERARCHY

We now study the dependence of the expressiveness of RelNNs on depth $D$.

RelNNs are generally defined to have a fixed depth, but allowing them to have a depth that is dependent on the number of nodes $n = |V|$ in the graph can substantially increase their expressive power. In the following, we define a *depth hierarchy* by analogy to the *time hierarchy* in computational complexity theory (Hartmanis & Stearns, 1965), and we extend our notation to let RelNN$[O(f(n)), B]$ denote the class of adaptive-depth RelNNs in which the growth-rate of depth $D$ is bounded by $O(f(n))$.

**Conjecture 3.3** (Depth hierarchy). For any maximum arity $B$, for any two functions $f$ and $g$, if $g(n) = o(f(n)/\log n)$, that is, $f$ grows logarithmically more quickly than $g$, then fixed-precision RelNN$[O(f(n)), B]$ is more expressive than fixed-precision RelNN$[O(g(n)), B]$.

There is a closely related result for the *congested clique* model in distributed computing, where Korhonen & Suomela (2018) proved that CLIQUE$(g(n)) \subsetneq$ CLIQUE$(f(n))$ if $g(n) = o(f(n))$. This result does not have the $\log n$ gap because the congested clique model allows $\log n$ bits to transmit between nodes at each iteration, while fixed-precision RelNN allows only a constant number of bits. The reason why the result on congested clique can not be applied to fixed-precision RelNNs is that congested clique assumes unbounded precision representation for each individual node.

However, Conjecture 3.3 is not true for RelNNs with unbounded precision, because there is an upper bound depth $O(n^{B-1})$ for a model's expressiveness power. That is, an unbounded-precision RelNN can not achieve stronger expressiveness by increasing its depth beyond $O(n^{B-1})$.

**Theorem 3.4** (Upper depth bound for unbounded-precision RelNN). For any maximum arity $B$, RelNN$[O(n^{B-1}), B] \succcurlyeq$ RelNN$[O(f(n)), B]$ for any function $f(n)$.

*Proof sketch.* Theorem 3.4 can be proved by combining the fact that the $B$-dimensional WL test is equivalent to differentiating graphs using FOC$_{B+1}$ (Cai et al., 1992), and the $O(n^B)$ upper bound on the computation depth for $B$-dimensional WL-test (Kiefer & Schweitzer, 2016). This upper

bound indicates that $O(n^B)$ depth is sufficient for $B$-dimensional WL-test to reach its maximum discriminative power, and thus any two graphs distinguishable by the WL-test can be distinguished by some $FOL_{B+1}$ formula with quantifier depth $O(n^B)$. The full proof is described in Appendix B.5.

It is important to point out that, to realize a specific graph reasoning function, RelNNs with different maximum arity $B$ may require different depth $D$. Fürer (2001) provides a general construction for problems that higher-dimensional RelNNs can solve in asymptotically smaller depth than lower-dimensional RelNNs. In the following we give a concrete example for computing *S-T Connectivity-k*, which asks whether there is a path of from nodes $S$ and $T$ in a graph, with length $\leq k$.

**Theorem 3.5** (S-T Connectivity-$k$ with Different Max Arity). For any function $f(k)$, if $f(k) = o(k)$, RelNN$[O(f(k)), 2]$ cannot realize S-T Connectivity-$k$. That is, S-T Connectivity-$k$ requires depth at least $O(k)$ for a relational neural network with an maximum arity of $B = 2$. However, S-T Connectivity-$k$ *can* be realized by RelNN$[O(\log k), 3]$.

*Proof sketch.* For any integer $k$, we can construct a graph with two chains of length $k$, so that if we mark two of the four ends as $S$ or $T$, any RelNN$[k - 1, 2]$ cannot tell whether $S$ and $T$ are on the same chain. The full proof is described in Appendix A.7.

There are many important graph reasoning tasks that do not have known depth lower bounds, including all-pair connectivity and shortest distance (Karchmer & Wigderson, 1990; Pai & Pemmaraju, 2019).

## 4 LEARNING AND GENERALIZATION IN RELATIONAL NEURAL NETWORKS

Given our understanding of what functions can be realized by relational neural networks, we move on to the problems of learning them: Can we effectively learn a relational neural network to solve a desired task given a sufficient number of input-output examples? What is the sample complexity? How do they generalize? These questions are very difficult to address in full for RelNNs, but there still some interesting things to be said. In this section we state and prove two theorems that illuminate their learning and generalization properties.

In a fixed-precision setting, we can show that applying *enumerative training* with examples up to some fixed size can ensure that the trained neural network will generalize to all graphs *larger* than those appearing in the training set. In an unbounded-precision setting, we can show that, if we use gradient descent to train models that use certain aggregation functions from randomly sampled data, then with a number of training examples that scales with $\log(N)$, the learned RelNN will generalize to all hypergraphs of size *smaller* than $N$.

### 4.1 GENERALIZATION OF FIXED-PRECISION RELATIONAL NEURAL NETWORKS

Machine learning studies of generalization focus on cases where input and output space are fixed-dimensional, and on how well the learned hypothesis can predict outputs for inputs that were not observed in the training set. Although learning specific graph reasoning functions in RelNNs are susceptible to this same type of analysis, we focus on an important type of generalization ability that is not present in the classical models: the ability to train on example graphs with up to $N$ nodes, and then generalize to make correct predictions on graphs with larger numbers of nodes.

A critical determinant of the generalization ability for RelNNs is the aggregation function they use. Specifically, Xu et al. (2019) have shown that using *sum* as the aggregation function provides maximum expressiveness for graph neural networks. However, sum aggregation cannot be implemented in fixed-precision models with an arbitrary number of nodes, because as the graph size $n$ increases, the range of the sum aggregation also increases.

**Definition 4.1** (Fixed-precision aggregation function). An aggregation function is *fixed precision* if it maps from any finite *set* of inputs with values drawn from *finite domains* to a *fixed finite* set of possible output values; that is, the cardinality of the range of the function cannot grow with the number of elements in the input set.

Two useful fixed-precision aggregation functions are *max*, which computes the dimension-wise maximum over the set of input values, and *fixed-precision mean*, which approximates the dimension-wise mean to a fixed decimal place.

**Definition 4.2** (Fixed-precision RelNN). A RelNN is *fixed precision* if its aggregation function is fixed precision and all of the component MLPs, $NN_j$, are also fixed precision, in the same sense as for aggregation functions above. This guarantees that the cardinality of the range of the hypergraph representation functions cannot grow with the number of nodes in the hypergraph.

In order to focus on structural generalization in this section, we consider an *enumerative* training paradigm. When the input hypergraph representation domain $\mathcal{S}$ is a finite set, we can enumerate the set $\mathcal{G}_{\leq N}$ of all possible input hypergraph representations of size bounded by $N$. We first enumerate all graph sizes $n \leq N$; for each $n$, we enumerate all possible values assigned to the hyperedges in the input. Given training size $N$, we enumerate all inputs in $\mathcal{G}_{\leq N}$, associate with each one the corresponding ground-truth output representation, and train the model with these input-output pairs.

This has much stronger data requirements than the standard sampling-based training mechanisms in machine learning. In the simplest binary graph classification case, enumerative training allows us to identify the optimal model in the hypothesis space. In practice, this can be approximated well when the input domain $\mathcal{S}$ is small and the input data distribution is approximately uniformly distributed. The enumerative learning setting is studied by the *language identification in the limit* community (Gold, 1967), in which it is called *complete presentation*. This is an interesting learning setting because even if the domain for each individual hyperedge representation is finite, as the graph size can go arbitrarily large, the number of possible inputs is enumerable but unbounded.

**Theorem 4.1** (Fixed-precision generalization under complete presentation). For any hypergraph reasoning function $f$, if it can be realized by a fixed-precision relational neural network model $\mathcal{M}$, then there exists an integer $N$, such that if we train the model with complete presentation on all input hypergraph representations with size smaller than $N$, $\mathcal{G}_{\leq N}$, then for all $M \in \mathcal{M}$,

$$\sum_{G \in \mathcal{G}_{\leq N}} 1[M(G) \neq f(G)] = 0 \implies \forall G \in \mathcal{G}_{\infty} : M(G) = f(G).$$

That is, as long as $M$ fits all training examples, it will generalize to all possible hypergraphs in $\mathcal{G}_{\infty}$.

*Proof.* The key observation is that for any fixed vector representation length $W$, there are only finite number of distinctive models in a fixed-precision RelNN family, *independent of the graph size $n$*. Let $W_b$ be the number of bits in each intermediate representation of a fixed-precision RelNN. There are at most $(2^{W_b})^{2^{W_b}}$ different mappings from inputs to outputs. Hence, if $N$ is sufficiently large to enumerate all input hypergraphs, we can always identify the correct model in the hypothesis space.

In future, we hope to extend this result to encompass sampling-based rather than enumerative training. In addition, models approximating the finite-precision assumption with sigmoid activation and max aggregation have empirically demonstrated the strong generalization properties this theorem predicts.

### 4.2 GENERALIZATION OF UNBOUNDED-PRECISION RELATIONAL NEURAL NETWORKS

Intuitively, relational neural networks with unbounded precision may not generalize to larger graphs because an aggregation function such as sum may accumulate through the layers. However, interestingly, this error can remain reasonably well-bounded for max or mean aggregation function. In this section, we will derive a bound for sample complexity in a more standard PAC-learning setting, in which there are no precision bounds on the inputs and computation of the model. It is difficult to characterize the VC dimension of an entire RelNN, but following Xu et al. (2021), we characterize the properties of a sequential training paradigm that considers the components independently.

**Theorem 4.2** (PAC sample complexity of unbounded-precision RelNNs). Let $\mathcal{M}$ be a family of unbounded-precision relational neural networks with hidden dimension $W$, depth $D$, and maximum arity $B$. Let $f$ be a graph function that can be realized by $\mathcal{M}$, and $M \in \mathcal{M}$ be a sequentially-trained model. If the following two conditions are true:
- (a) There exists an instance $M' \in \mathcal{M}$ such that $M'$ realizes $f$ and MLPs in $M$ are $(\epsilon, \delta)$-approximations of those in $M'$ [*]; and
- (b) The MLPs $\text{NN}_j$ in $M$ are Lipschitz-continuous i.e. $\|\text{NN}_j(x_1) - \text{NN}_j(x_2)\| \leq \lambda \|x_1 - x_2\|$ for some constant $\lambda$ and for all $j$.

Then on graphs of size $n$, $M$ is a $\left(O\left(\epsilon \lambda^D W^{D/2}\right), O(\delta D n^B)\right)$-approximation of $f$ if $\mathcal{M}$ uses max aggregation, and a $\left(O\left(\epsilon \lambda^D\right), O(\delta D n^B)\right)$-approximation of $f$ if $\mathcal{M}$ uses mean aggregation.

Theorem 4.2 indicates that when using max or mean aggregation, the generalization error does not scale with the graph size $n$ (which is the case for *sum*), while the failure probability $\delta$ scales with $O(Dn^B)$. In the PAC-learning framework, the sample complexity for achieving error $\epsilon$ with failure probability $\delta$ is bounded by $O\left(\frac{1}{\epsilon}\left(\log\left(VC(\mathcal{H})\right) + \log\left(\frac{1}{\delta}\right)\right)\right)$, where $VC(\mathcal{H})$ is the VC-dimension

---

[*] That is, $\Pr_{x \sim \mathcal{X}}(\|M(x) - M'(x)\| \leq \epsilon) \geq (1 - \delta)$, where $\mathcal{X}$ is the data distribution

| Model | Agg. | 3-link | | 4-link | | triangle | | 4-clique | |
|---|---|---|---|---|---|---|---|---|---|
| | | $n = 10$ | $n = 30$ | $n = 10$ | $n = 30$ | $n = 10$ | $n = 30$ | $n = 10$ | $n = 30$ |
| 1-ary GNN | Max | $70.0_{\pm 0.0}$ | $82.7_{\pm 0.0}$ | $92.0_{\pm 0.0}$ | $91.7_{\pm 0.0}$ | $73.7_{\pm 3.2}$ | $50.2_{\pm 1.8}$ | $55.3_{\pm 4.0}$ | $46.2_{\pm 1.3}$ |
| | Sum | $100.0_{\pm 0.0}$ | $89.4_{\pm 0.4}$ | $100.0_{\pm 0.0}$ | $86.1_{\pm 1.2}$ | $77.7_{\pm 8.5}$ | $48.6_{\pm 1.6}$ | $53.7_{\pm 0.6}$ | $55.2_{\pm 0.8}$ |
| 2-ary NLM | Max | $65.3_{\pm 0.6}$ | $54.0_{\pm 0.6}$ | $93.0_{\pm 0.0}$ | $95.7_{\pm 0.0}$ | $51.0_{\pm 1.7}$ | $49.2_{\pm 0.4}$ | $55.0_{\pm 0.0}$ | $45.7_{\pm 0.0}$ |
| | Sum | $100.0_{\pm 0.0}$ | $88.3_{\pm 0.0}$ | $100.0_{\pm 0.0}$ | $67.4_{\pm 16.4}$ | $82.0_{\pm 2.6}$ | $48.3_{\pm 0.0}$ | $53.0_{\pm 0.0}$ | $54.4_{\pm 1.5}$ |
| 2-ary GNN | Max | $78.7_{\pm 0.6}$ | $76.0_{\pm 17.3}$ | $97.7_{\pm 4.0}$ | $98.6_{\pm 2.5}$ | $100.0_{\pm 0.0}$ | $100.0_{\pm 0.0}$ | $55.0_{\pm 0.0}$ | $45.7_{\pm 0.0}$ |
| | Sum | $100.0_{\pm 0.0}$ | $51.2_{\pm 7.9}$ | $100.0_{\pm 0.0}$ | $45.7_{\pm 7.6}$ | $100.0_{\pm 0.0}$ | $49.2_{\pm 1.0}$ | $61.0_{\pm 5.6}$ | $54.3_{\pm 0.0}$ |
| 3-ary NLM | Max | $100.0_{\pm 0.0}$ | $100.0_{\pm 0.0}$ | $100.0_{\pm 0.0}$ | $100.0_{\pm 0.0}$ | $100.0_{\pm 0.0}$ | $100.0_{\pm 0.0}$ | $59.0_{\pm 6.9}$ | $45.9_{\pm 0.4}$ |
| | Sum | $100.0_{\pm 0.0}$ | $87.6_{\pm 11.0}$ | $100.0_{\pm 0.0}$ | $65.4_{\pm 14.3}$ | $100.0_{\pm 0.0}$ | $80.6_{\pm 8.8}$ | $73.7_{\pm 13.8}$ | $53.3_{\pm 8.8}$ |
| 3-ary GNN | Max | $79.0_{\pm 0.0}$ | $86.0_{\pm 0.0}$ | $100.0_{\pm 0.0}$ | $100.0_{\pm 0.0}$ | $100.0_{\pm 0.0}$ | $100.0_{\pm 0.0}$ | $84.0_{\pm 0.0}$ | $93.3_{\pm 0.0}$ |
| | Sum | $100.0_{\pm 0.0}$ | $84.1_{\pm 18.6}$ | $100.0_{\pm 0.0}$ | $61.1_{\pm 15.0}$ | $100.0_{\pm 0.0}$ | $95.1_{\pm 7.3}$ | $80.5_{\pm 0.7}$ | $66.2_{\pm 19.6}$ |
| 4-ary NLM | Max | $100.0_{\pm 0.0}$ | $100.0_{\pm 0.0}$ | $100.0_{\pm 0.0}$ | $100.0_{\pm 0.0}$ | $100.0_{\pm 0.0}$ | $100.0_{\pm 0.0}$ | $82.0_{\pm 1.7}$ | $93.1_{\pm 0.2}$ |
| | Sum | $100.0_{\pm 0.0}$ | $59.1_{\pm 5.3}$ | $100.0_{\pm 0.0}$ | $67.7_{\pm 24.1}$ | $100.0_{\pm 0.0}$ | $82.1_{\pm 12.8}$ | $84.0_{\pm 0.0}$ | $67.0_{\pm 18.9}$ |

Table 2: Overall accuracy on relational reasoning problems. All models are trained on $n = 10$, and tested on $n = 30$. The standard error of all values are computed based on three random seeds.

of the hypothesis space. Hence, scaling to graphs with size $n$ requires additional $O(B \log n)$ samples when considering $W, D, \lambda$ as constants. The proof of Theorem 4.2 can be found in Appendix B.7.

The sample complexity does not depend on the graph size, because under the *sequential learning* setting the MLPs are trained with direct supervision. Though this setting can not be implemented in practice, in our experiments we observe good generalization of models trained end-to-end.

Theorem 4.2 is interesting because it shows that using max or mean aggregation has provably better sample complexity than using sum aggregation. Combined with the fact that using max aggregation, RelNN$[\cdot, B]$ can realize FOC$_B$ with only $\exists$ and $\forall$ quantifiers (Dong et al., 2019), RelNN with max aggregation is a strong model class.

## 5 EXPERIMENTS

We now study how our theoretical results on model expressiveness and learning apply to relational neural networks trained with gradient descent on practically meaningful problems. We begin by describing two synthetic benchmarks: graph substructure detection and relational reasoning.

In the graph substructure detection dataset, there are several tasks of predicting whether there input graph contain a sub-graph with specific structure. The tasks are: *3-link* (length-3 path), *4-link*, *triangle*, and *4-clique*. These are important graph properties with many potential applications.

The relational reasoning dataset is composed of two family-relationship prediction tasks and two connectivity-prediction tasks. They are all *binary edge classification* tasks. In the family-relationship prediction task, the input contains the *mother* and *father* relationships, and the task is to predict the *grandparent* and *uncle* relationships between all pairs of entities. In the connectivity-prediction tasks, the input is the edges in an undirected graph and the task is to predict, for all pairs of nodes, whether they are connected with a path of length $\leq 4$ (*connectivity-4*) and whether they are connected with a path of arbitrary length (*connectivity*). The data generation for all datasets is included in Appendix C.

### 5.1 RESULTS

Our main results on all datasets are shown in Table 2 and Table 3. We empirically compare relational neural networks with different maximum arity $B$, different model architecture (GNN and NLM), and different aggregation functions (max and sum). All models use sigmoidal activation for all MLPs. For each task on both datasets we train on a set of small graphs ($n = 10$) and test the trained model on both small graphs and large graphs ($n = 10$ and $n = 30$). We summarize the findings below.

**Expressiveness.** We have seen a theoretical equal expressiveness between GNNs and NLMs applied to hypergraphs. That is, a GNN applied to $B$-ary hyperedges is equivalent to a $(B + 1)$-ary NLM. Table 2 and 3 further suggest their similar performance on tasks when trained with gradient descent.

Formally, triangle detection requires RelNNs with at least $B = 3$ to solve. Thus, we see that all RelNNs with arity $B = 2$ fail on this task, but models with $B = 3$ perform well. Formally, 4-clique is realizable by RelNNs with maximum arity $B = 4$, but we failed to reliably train models to reach perfect accuracy on this problem. It is not yet clear what the cause of this behavior is.

| Model | Agg. | grand parent | | uncle | | connectivity-4[†] | | connectivity | |
|---|---|---|---|---|---|---|---|---|---|
| | | $n=20$ | $n=80$ | $n=20$ | $n=80$ | $n=10$ | $n=80$ | $n=10$ | $n=80$ |
| 1-ary GNN | Max | $84.0_{\pm0.3}$ | $64.8_{\pm0.0}$ | $93.6_{\pm0.3}$ | $66.1_{\pm0.0}$ | $72.6_{\pm3.6}$ | $67.5_{\pm0.5}$ | $85.6_{\pm0.3}$ | $75.1_{\pm1.9}$ |
| | Sum | $84.7_{\pm0.1}$ | $64.4_{\pm0.0}$ | $94.3_{\pm0.2}$ | $66.2_{\pm0.0}$ | $79.6_{\pm0.1}$ | $68.3_{\pm0.1}$ | $87.1_{\pm0.3}$ | $75.0_{\pm0.2}$ |
| 2-ary NLM | Max | $82.3_{\pm0.5}$ | $65.6_{\pm0.1}$ | $93.1_{\pm0.0}$ | $66.6_{\pm0.0}$ | $91.2_{\pm0.2}$ | $51.0_{\pm0.6}$ | $88.9_{\pm2.6}$ | $67.1_{\pm4.8}$ |
| | Sum | $82.9_{\pm0.1}$ | $64.6_{\pm0.1}$ | $93.4_{\pm0.0}$ | $66.7_{\pm0.2}$ | $96.0_{\pm0.4}$ | $68.3_{\pm0.5}$ | $84.0_{\pm0.0}$ | $71.9_{\pm0.0}$ |
| 2-ary GNN | Max | $100.0_{\pm0.0}$ | $100.0_{\pm0.0}$ | $100.0_{\pm0.0}$ | $100.0_{\pm0.0}$ | $100.0_{\pm0.0}$ | $100.0_{\pm0.0}$ | $84.0_{\pm0.0}$ | $71.9_{\pm0.0}$ |
| | Sum | $100.0_{\pm0.0}$ | $35.7_{\pm0.0}$ | $100.0_{\pm0.0}$ | $33.9_{\pm0.0}$ | $100.0_{\pm0.0}$ | $51.3_{\pm5.3}$ | $84.0_{\pm0.0}$ | $71.9_{\pm0.0}$ |
| 3-ary NLM | Max | $100.0_{\pm0.0}$ | $100.0_{\pm0.0}$ | $100.0_{\pm0.0}$ | $100.0_{\pm0.0}$ | $100.0_{\pm0.0}$ | $100.0_{\pm0.0}$ | $100.0_{\pm0.0}$ | $100.0_{\pm0.1}$ |
| | Sum | $100.0_{\pm0.0}$ | $35.7_{\pm0.0}$ | $100.0_{\pm0.0}$ | $50.8_{\pm29.4}$ | $100.0_{\pm0.0}$ | $77.8_{\pm11.8}$ | $100.0_{\pm0.0}$ | $88.2_{\pm8.0}$ |
| 3-ary NLM$_{HE}$ | Max | $100.0_{\pm0.0}$ | $100.0_{\pm0.0}$ | $100.0_{\pm0.0}$ | $100.0_{\pm0.0}$ | N/A | N/A | N/A | N/A |
| | Sum | $100.0_{\pm0.0}$ | $35.7_{\pm0.0}$ | $100.0_{\pm0.0}$ | $33.8_{\pm29.4}$ | N/A | N/A | N/A | N/A |

Table 3: Overall accuracy on relational reasoning problems. Models for family-relationship prediction are trained on $n=20$, while models for connectivity problems are trained on $n=10$. All model are tested on $n=80$. The standard error of all values are computed based on three random seeds. The 3-ary NLMs marked with "HE" have hyperedges in inputs, where each family is represented by a 3-ary hyperedge instead of two parent-child edges, and the results are similar to binary edges.

**Structural generalization.** We discussed the structural generalization properties of RelNNs in Section 4.1, in a learning setting based on fixed-precision networks and enumerative training. This setting can be *approximated* by training RelNNs with max aggregation and sigmoidal activation on sufficient data. In both Table 2 and Table 3 we see that, when models can realize the desired function and are trained to 100% accuracy on the training split, they also generalize well to larger graphs. In appendix C, we provide a more detailed study on the generalization performance.

**Sample complexity (case study).** In Section 4.2, we discussed the sampling complexity differences induced by different aggregation functions in the *sequential learning* setting. The primary conclusion is that models with *max* aggregation are expected to have better sample complexity than models with *sum* aggregation. We test this hypothesis in the *end-to-end* learning setting. For the triangle detection problem, we train 3D-NLM with sum and max aggregation with fewer training samples. We observe that sum aggregation requires about 300 training samples to reach nearly perfect accuracy, while max aggregation requires only 30 training samples. More details are included in appendix C.

## 6 RELATED WORK

Solving problems on graphs of arbitrary size is studied in many fields. RelNNs can be viewed as circuit families with constrained architecture. In distributed computation, the congested clique model can be viewed as 2-arity RelNNs, where nodes have identities as extra information. Common graph problems including sub-structure detection(Li et al., 2017; Rossman, 2010) and connectivity(Karchmer & Wigderson, 1990) are studied for lower bounds in terms of depth, width and communication. This has been connected to GNNs for deriving expressiveness bounds (Loukas, 2020).

Studies have been conducted on the expressiveness of GNNs and their variants. Xu et al. (2019) provide an illuminating characterization of GNN expressiveness in terms of the WL graph isomorphism test. Barceló et al. (2020) reviewed GNNs from the logical perspective and rigorously refined their logical expressiveness with respect to fragments of first-order logic. Dong et al. (2019) proposed Neural Logical Machines (NLMs) to reason about higher-order relations, and showed that increasing order inreases expressiveness. It is also possible to gain expressiveness using unbounded computation time, as shown by the work of Dehghani et al. (2019) on dynamic halting in transformers.

It is interesting that GNNs may generalize to larger graphs. Xu et al. (2020; 2021) have studied the notion of *algorithmic alignment* to quantify such structural generalization. Dong et al. (2019) provided empirical results showing that NLMs generalize to much larger graphs on certain tasks.

## 7 CONCLUSION

We have provided a framework for unifying several powerful classes of relational neural networks and characterized aspects of their representational and generalization power, both theoretically and empirically. In particular, we have shown the substantial increase of expressive power due to higher-arity relations and increasing depth, and have characterized very powerful structural generalization from training on small graphs to performance on larger ones. Although many questions remain open about the overall generalization capacity of these models in continuous and noisy domains, we believe this work has shed some light on their utility and potential for application in a variety of problems.

**Reproducibility statement.** The proof details for theorems in this paper have been included in the appendix.

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

# A    GRAPH PROBLEMS

We listed a number of examples for graph classification and regression tasks, and we provide the definitions and the current best known RelNNs for learning these tasks from data. For some of the problems, we will also show why they can not be solved by a simpler problems, or indicate them as open problems.

## A.1    NODE COLOR MAJORITY

Each node is assigned a color $c \in \mathcal{C}$ where $\mathcal{C}$ is a finite set of all colors. The model needs to predict which color the most nodes have.

Using a single layer with sum aggregation, the model can count the number of nodes of color $c$ for each $c \in \mathcal{C}$ on its global representation.

## A.2    COUNT RED NODES

Each node is assigned a color of red or blue. The model needs to count the number of red nodes.

Similarly, using a single layer with sum aggregation, the model can count the number of red nodes on its global representation.

## A.3    3-LINK DETECTION

Given an unweighted, undirected graph, the model needs to detect whether there is a triple of nodes $(a, b, c)$ such that $a \neq c$ and $(a, b)$ and $(b, c)$ are edges.

This is equivalent to check whether there exists a node with degree at least 2. We can use a Reduction operation with sum aggregation to compute the degree for each node, and then use a Reduction operation with max aggregation to check whether the maximum degree of nodes is greater than or equal to 2.

Note that this can not be done with 1 layer, because the edge information is necessary for the problem, and they require at least 2 layers to be passed to the global representation.

## A.4    4-LINK DETECTION

Given an unweighted undirected graph, the model needs to detect whether there is a 4-tuple of nodes $(a, b, c, d)$ such that $a \neq c, b \neq d$ and $(a, b), (b, c), (c, d)$ are edges (note that a triangle is also a 4-link).

This problem is equivalent to check whether there is an edge between two nodes with degrees $\geq 2$. We can first reduce the edge information to compute the degree for each node, and then expand it back to 2-dimensional representations, so we can check for each edge if the degrees of its ends are $\geq 2$. Then the results are reduced to the global representation with existential quantifier (realized by max aggregation) in 2 layers.

## A.5    TRIANGLE DETECTION

Given a unweighted undirected graph, the model is asked to determine whether there is a triangle in the graph i.e. a tuple $(a, b, c)$ so that $(a, b), (b, c), (c, a)$ are all edges.

This problem can be solved by RelNN [4,3]: we first expand the edge to 3-dimensional representations, and determine for each 3-tuple if they form a triangle. The results of 3-tuples require 3 layers to be passed to the global representation.

We can prove that Triangle Detection indeed requires breadth at least 3. Let $k$-regular graphs be graphs where each node has degree $k$. Consider two $k$-regular graphs both with $n$ nodes, so that exactly one of them contains a triangle[‡]. However, RelNNs of breadth 2 has been proven not to be stronger than WL test on distinguish graphs, and thus can not distinguish these two graphs (WL test can not distinguish any two $k$-regular graphs with equal size).

## A.6    4-CLIQUE DETECTION AND COUNTING

Given an undirected graph, check existence of, or count the number of tuples $(a, b, c, d)$ so that there are edges between every pair of nodes in the tuple.

---

[‡]Such construction is common. One example is $k = 2, n = 6$, and the graph may consist of two separated triangles or one hexagon

This problem can be easily solved by a RelNN with breadth 4 that first expand the edge information to the 4-dimensional representations, and for each tuple determine whether its is a 4-clique. Then the information of all 4-tuples are reduced 4 times to the global representation (sum aggregation can be used for counting those).

Though we did not find explicit counter-example construction on detecting 4-cliques with RelNNs of breadth 3, we suggest that this problem can not be solved with RelNNs with 3 or lower breadth.

## A.7 CONNECTIVITY

The connectivity problems are defined on unweighted undirected graphs. S-T connectivity problems provides two nodes $S$ and $T$ (labeled with specific colors), and the model needs to predict if they are connected by some edges. All pair connectivity problem require the model to answer for every pair of nodes. Connectivity-$k$ problems have an additional requirement that the distance between the pair of nodes can not exceed $k$.

S-T connectivity-$k$ can be solved by a RelNN of breadth 2 with $k$ iterations. Assume $S$ is colored with color $c$, at every iteration, every node with color $c$ will spread the color to its neighbors. Then, after $k$ iterations, it is sufficient to check whether $T$ has the color $c$.

With RelNNs of breadth 3, we can use $O(\log k)$ matrix multiplications to solve connectivity-$k$ between every pair of nodes. Since the matrix multiplication can naturally be realized by RelNNs of breadth 3 with two layers. All-pair connectivity problems can all be solved with $O(\log k)$ layers.

**Theorem A.1** (S-T connectivity-$k$ with RelNN[$o(k)$, 2]). S-T connectivity-$k$ can not be solved by a RelNN of maximum arity within $o(k)$ iterations.

*Proof.* We construct two graphs each has $2k$ nodes $u_1, \cdots, u_k, v_1, \cdots, v_k$. In both graph, there are edges $(u_i, u_{i+1})$ and $(v_i, v_{i+1})$ for $1 \leq i \leq k - 1$ i.e. there are two links of length $k$. We then set $S = u_1, T = u_n$ and $S = u_1, T = v_n$ the the two graphs.

We will analysis GNNs as RelNNs are proved to be equivalent to them by scaling the depth by a constant factor. Now consider the node refinement process where each node $x$ is refined by the multiset of labels of $x$'s neighbots and the multiiset of labels of $x$'s non-neighbors.

Let $C_j^{(i)}(x)$ be the label of $x$ in graph $j$ after $i$ iterations, at the beginning, WLOG, we have

$$C_1^{(0)}(u_1) = 1, C_1^{(0)}(u_n) = 2 C_1^{(0)}(u_1) = 1, C_1^{(0)}(v_n) = 2$$

and all other nodes are labeled as 0.

Then we can prove by induction: after $i \leq \frac{k}{2} - 1$ iterations, for $1 \leq t \leq i + 1$ we have

$$C_1^{(u_t)} = C_2^{(i)}(u_t), C_1^{(v_t)} = C_2^{(i)}(v_t)$$

$$C_1^{(u_{k-t+1})} = C_2^{(i)}(v_{k-t+1}), C_1^{(v_{k-t+1})} = C_2^{(i)}(u_{k-t+1})$$

and for $i + 2 \leq t \leq k - i - 1$ we have

$$C_1^{(u_t)} = C_2^{(i)}(u_t), C_1^{(v_t)} = C_2^{(i)}(v_t)$$

This is true because before $\frac{k}{2}$ iterations are run, the multiset of all node labels are identical for the two graphs (say $S^{(i)}$). Hence each node $x$ is actually refined by its neighbors and $S^{(i)}$ where $S^{(i)}$ is the same for all nodes. Hence, before running $\frac{k}{2}$ iterations when the message between $S$ and $T$ finally meets in the first graph, GNN can not distinguish the two graphs, and thus can not solve the connectivity with distance $k - 1$. □

## A.8 MAX DEGREE

The max degree problem gives a graph and ask the model to output the maximum degree of its nodes.

Like we mentioned in 3-link detection, one layer for computing the degree for each node, and another layer for taking the max operation over nodes should be sufficient.

### A.9 MAX FLOW

The Max Flow problem gives a directional graph with capacities on edges, and indicate two nodes $S$ and $T$. The models is then asked to compute the amount of max-flow from $S$ to $T$.

Notice that the Breadth First Search (BFS) component in Dinic's algorithm(Dinic, 1970) can be implemented on RelNNs as they does not require node identities (all new-visited nodes can augment to their non-visited neighbors in parallel). Since the BFS runs for $O(n)$ iteration, and the Dinic's algorithm runs BFS $O(n^2)$ times, the max-flow can be solved by RelNNs with in $O(n^3)$ iterations.

### A.10 DISTANCE

Given a graph with weighted edges, compute the length of the shortest between specified node pair (S-T Distance) or all node pairs (All-pair Distance).

Similar to Connectivity problems, but Distance problems now additionally record the minimum distance from $S$ (for S-T) or between every node pairs (for All-pair), which can be updated using min operator (using Min-plus matrix multiplication for All-pair case).

## B PROOFS AND ANALYSIS

### B.1 CONSTANT-LAYER RELNNS AND RECURRENT RELNNS

**Theorem B.1.** A neural network with representation width $W$ that has $D$ different layers $\mathrm{NN}_1, \cdots, \mathrm{NN}_D$ can be realized by a neural network that applies a single layer $\mathrm{NN}'$ for $D$ iterations with width $(D+1)(W+1)$.

*Proof.* The representation for $\mathrm{NN}'$ can be partitioned into $D+1$ segments each of length $W+1$. Each segment consist of a "flag" element and a $W$-element representation, which are all $0$ initially, except for the first segment, where the flag is set to $1$, and the representation is the input.

$\mathrm{NN}'$ has the weights for all $\mathrm{NN}_1, \cdots, \mathrm{NN}_D$, where weights $\mathrm{NN}_i$ are used to compute the representation in segment $i+1$ from the representation in segment $i$. Additionally, at each iteration, segment $i+1$ can only be computed if the flag in segment $i$ is $1$, in which case the flag of segment $i+1$ is set to $1$. Clearly, after $D$ iterations, the output of $\mathrm{NN}_k$ should be the representation in segment $D+1$. □

Due to Theorem B.1, we consider the neural networks that recurrently apply the same layer because a) they are as expressive as those using layers of different weights, b) it is easier to analyze a single neural network layer than $D$ layers, and c) they naturally generalize to neural networks that runs for adaptive number of iterations (e.g. GNNs that run $O(\log n)$ iterations where $n$ is the size of the input graph).

### B.2 EXPRESSIVENESS EQUIVALENCE OF RELATIONAL NEURAL NETWORKS

We first describe a framework for quantifying if two RelNN models are equally expressive on regression tasks. The framework view the expressiveness from the perspective of computation. Specifically, we will prove the expressiveness equivalence between models by showing that their computation can be aligned.

In complexity, we usually show a problem is at least as hard as the other one by showing a reduction from the other problem to the problem. Similarly, on the expressiveness of RelNNs, we can construct reduction from model family $\mathcal{A}$ to model family $\mathcal{B}$ to show that $\mathcal{B}$ can realize all computation that $\mathcal{A}$ does, or even more. Formally, we have the following definition.

**Definition B.1** (Expressiveness reduction). For two model families $\mathcal{A}$ and $\mathcal{B}$, we say $\mathcal{A}$ can be **reduced** to $\mathcal{B}$ if and only if there is a function $r : \mathcal{A} \to \mathcal{B}$ such that for each model instance $A \in \mathcal{A}$, $r(A) \in \mathcal{B}$ and $A$ have the same outputs on all inputs. In this case, we say $\mathcal{B}$ is at least as expressive as $\mathcal{A}$.

**Definition B.2** (Expressiveness equivalence). For two model families $\mathcal{A}$ and $\mathcal{B}$, if $\mathcal{A}$ and $\mathcal{B}$ can be reduced to each other, then $\mathcal{A}$ and $\mathcal{B}$ are equally expressive. Note that this definition of expressiveness equivalence generalizes to both classification and regression tasks.

**Graph Neural Networks.** GNNs (Scarselli et al., 2008) is defined based on two message passing operations.

- Edge update: the feature of each edge is updated by features of its ends.
- Note update: the feature of each node is updated by features of all edges adjacent to it.

This message passing scheme can easily fit in RelNN architecture by storing edge features as tuples of two nodes. In this case, updating edges by its ends can be realized by the *expand* and the *permute* operation. The message aggregation at each node can be realized by a *reduce* operation. Hence, GNNs can be realized by a RelNN of maximum arity $B = 2$.

**High-dimensional GNNs.** Computing only node-wise and edge-wise features does not handle higher-order relations, such as triangles in the graph. In order to obtain more expressive power, GNNs have be extend to hypergraphs of hier arity (Morris et al., 2019). Specifically, GNNs on $B$-ary hypergraph maintains features for all $B$-tuple of nodes, and the neighborhood is extended to $B$-tuples accordingly: the feature of tuple $(v_1, v_2, \cdots, v_B)$ is updated by the $|V|$ element multiset (contain $|V|$ elements for each $u \in V$) of $B$-tuples of features

$$(H_i[u, v_2, \cdots, v_B], H_{i-1}[v_1, u, v_2, \cdots, v_B], \cdots H_{i-1}[v_1, \cdots, v_{B-1}, u]) \tag{B.1}$$

where $H_{i-1}[\boldsymbol{v}]$ is the feature of tuple $\boldsymbol{v}$ from the previous iteration.

We now introduce the formal definition of the high-dimensional message passing. We denote $\boldsymbol{v}$ as a $B$-tuple of nodes $(v_1, v_2, \cdots, v_B)$, and generalize the neighborhood to a higher dimension by defining the neighborhood of $\boldsymbol{v}$ as all node tuples that differ from $\boldsymbol{v}$ at one position.

$$\text{Neighbors}(\boldsymbol{v}, u) = ((u, v_2, \cdots, v_B), (v_1, u, v_3, \cdots, v_B), \cdots, (v_1, \cdots, v_{B-1}, u)) \tag{B.2}$$
$$N(\boldsymbol{v}) = \{\text{Neighbors}(\boldsymbol{v}, u) | u \in V\} \tag{B.3}$$

Then message passing scheme naturally generalizes to high-dimensional features using the high-dimensional neighborhood.

$$\text{Received}_i[\boldsymbol{v}] = \sum_u \left( \text{NN}_1 \left( H_{i-1}[\boldsymbol{v}]; \text{CONCAT}_{\boldsymbol{v}' \in \text{neighbors}(\boldsymbol{v}, u)} H_{i-1}[\boldsymbol{v}'] \right) \right) \tag{B.4}$$

Recall that the normal GNNs can be realize by RelNN$[\cdot, 2]$. Similarly, since features of $B$-tuples can be stored in RelNNs with arity $B$ and the high-dimensional message passing involves at most $B + 1$ nodes at the same time, $(v_1, \cdots, v_B$ and $u)$, we can extend the previous result to that GNNs on $B$-arity hypergraphs can be realized by RelNNs with maximum arity $B + 1$.

**Equivalence between hypergraph GNNs and RelNNs.** We will prove the equivalence between hypergraph GNNs and RelNNs by making reductions in both directions.

**Lemma B.2.** A GNN on $B$-ary hypergraphs and depth $D$ can be realized by a RelNN with maximum arity $B + 1$ and depth $2D$.

*Proof.* We prove lemma B.2 by showing that one layer of GNNs on $B$-ary hypergraphs can be realized by two RelNN with maximum arity $B + 1$.

Firstly, a GNN layer maintain features of $B$-tuples, which are stored in correspondingly in an RelNN layer at dimension $B$. Then we will realize the message passing scheme using the RelNN features of dimension $B$ and $B + 1$ in two steps.

Recall the message passing scheme generalized to high dimensions (to distinguish, we use $H$ for GNN features and $T$ for RelNN features.)

$$\text{Received}_i(\boldsymbol{v}) = \sum_u \left( \text{NN}_1 \left( H_{i-1, B}[\boldsymbol{v}]; \text{CONCAT}_{\boldsymbol{v}' \in \text{neighbors}(\boldsymbol{v}, u)} H_{i-1}[\boldsymbol{v}'] \right) \right) \tag{B.5}$$

At the first step, the Expand operation first raise the dimension to $B + 1$ by expanding a non-related variable $u$ to the end, and the Permute operation can then swap $u$ with each of the elements (or no swap). Particularly, $T_{i,B}[v_1, v_2, \cdots, v_B]$ will be expand to

$$T_{i+1,B+1}[u, v_2, v_3, \cdots, v_B, v_1], T_{i+1,B+1}[v_1, u, v_3, \cdots, v_B, v_2], \cdots,$$
$$T_{i+1,B+1}[v_1, v_2, \cdots, v_{B-1}, u, v_B], \text{and } T_{i+1,B+1}[v_1, v_2, \cdots, v_{B-1}, v_B, u]$$

Hence, $T_{i+1,B+1}[v_1, v_2, v_3, \cdots, v_B, u]$ receives the features from

$$T_{i,B}[v_1, v_2, \cdots, v_B], T_{i,B}[u, v_2, v_3, \cdots, v_B], T_{i,B}[v_1, u, v_3, \cdots, v_B], \cdots, T_{i,B}[v_1, v_2, \cdots, v_{B-1}, u]$$

These features matches the input of $\mathrm{NN}_1$ in equation B.5, and in this layer $\mathrm{NN}_1$ can be applied to compute things inside the summation.

Then at the second step, the last element is reduced to get what tuple $\boldsymbol{v}$ should receive, so $\boldsymbol{v}$ can be updated. Since each GNN layer can be realized by such two RelNN layers, each GNN on $B$-ary hypergraphs with depth $D$ can be realized by a RelNN of maximum arity $(B + 1)$ and depth $2D$. $\quad\square$

To complete the proof we need to find a reduction from RelNNs of maximum arity $B + 1$ to GNNs on $B$-ary hypergraphs. The key observation here is that the features of $(B + 1)$-tuples in RelNNs can only be expanded from sub-tuples, and the expansion and reduction involving $(B + 1)$-tuples can be simulated by the message passing process.

**Lemma B.3.** The features of $(B + 1)$-tuples feature $T_{i,B+1}[v_1, v_2, \cdots, v_{B+1}]$ can be computed from the following tuples

$$\left(T_{i,B}[v_2, v_3, \cdots, v_{B+1}], T_{i,B}[v_1, v_3, \cdots, v_{B+1}], \cdots, T_{i,B}[v_1, v_2, \cdots, v_B]\right).$$

*Proof.* Lemma B.3 is true because $(B + 1)$-dimensional representations can either be computed from themselves at the previous iteration, or expanded from $B$-dimensional representations. Since representations at all previous iterations $j < i$ can be contained in $T_{i,B}$, it is sufficient to compute $T_{i,B+1}[v_1, v_2, \cdots, v_{B+1}]$ from all its $B$-ary sub-tuples. $\quad\square$

Then let's construct the GNN for given RelNN to show the existence of the reduction.

**Lemma B.4.** A RelNN of maximum arity $B + 1$ and depth $D$ can be realized by a GNN on $B$-ary hypergraphs with no more than $D$ iterations.

*Proof.* We can realize the Expand and Reduce operation with only the $B$-dimensional features using the broadcast message passing scheme. Note that Expand and Reduce between $B$-dimensional features and $(B + 1)$-dimensional features in the RelNN is a special case where claim B.3 is applied.

Let's start with Expand and Reduce operations between features of dimension $B$ or lower. For the $b$-dimensional feature in the RelNN, we keep $n^{\underline{b}} n^{B-b}$[§] copies of it and store them the representation of every $B$-tuple who has a sub-tuple[¶] that is a permutation of the $b$-tuple. That is, for each $B$-tuple in the GNN on $B$-ary hypergraphs, for its every sub-tuple of length $b$, we store $b!$ representations corresponding to every permutation of the $b$-tuple in the RelNN. Keeping representation for all sub-tuple permutations make it possible to realize the Permute operation. Also, it is easy to notice that Expand operation is realized already, as all features with dimension lower than $B$ are naturally expanded to $B$ dimension by filling in all possible combinations of the rest elements. Finally, the Reduce operation can be realized using a broadcast casting message passing on certain position of the tuple.

Now let's move to the special case – the Expand and Reduce operation between features of dimensions $B$ and $B + 1$. Claim B.3 suggests how the $(B + 1)$-dimensional features are stored in $B$-dimensional representations in GNNs, and we now show how the Reduce can be realized by message passing.

---

[§] $n^{\underline{k}} = n \times (n - 1) \times \cdots \times (n - k + 1)$.

[¶] The sub-tuple does not have to be consecutive, but instead can be a any subset of the tuple that keeps the element order.

We first bring in claim B.3 to the GNN message passing, where we have $\text{Received}_i[\boldsymbol{v}]$ to be

$$\sum_u \left( \text{NN}_1 \left( T_{i-1,B}[v_2, v_3, \cdots, v_B, u], T_{i-1,B}[v_1, v_3, \cdots, v_B, u], \cdots, T_{(i-1),B}[v_1, v_2, \cdots, v_B] \right) \right)$$

Note that the last term $T_{i-1,B}[v_1, v_2, \cdots, v_B]$ is contained in $H_{i-1}(v)$ in equation B.5, and other terms are contained in $H_{i-1}(v')$ for $v' \in \text{neighbors}(\boldsymbol{v}, u)$. Hence, equation B.5 is sufficient to simulate the Reduce operation. □

**Theorem B.5.** GNNs on $B$-ary hypergraphs are equally expressive as RelNNs with maximum arity $B + 1$.

*Proof.* This is a direct conclusion by combining Lemma B.2 and Lemma B.4. □

### B.3 EXPRESSIVENESS OF HYPERGRAPH CONVOLUTION AND ATTENTION

Hypergraph convolution(Feng et al., 2019; Yadati et al., 2019; Bai et al., 2021), attention(Ding et al., 2020) and message passing(Huang & Yang, 2021) focus on updating node features through hyperedges instead of hyperedges. These approaches can be viewed as instances of RelNNs, and they have smaller time complexity because they do not model all high-arity tuples. However, they are less expressive than RelNNs with equal max arity.

These approaches can be formulated to two steps at each iteration. At the first step, each hyperedge is updated by the features of nodes it connects.

$$h_{i,e} = \text{AGG}_{v \in e} f_{i-1,v} \tag{B.6}$$

At the second step, each node is updated by the features of hyperedges connecting it.

$$f_{i,v} = \text{AGG}_{v \in e} h_{i,e} \tag{B.7}$$

where $f_{i,v}$ is the feature of node $v$ at iteration $i$, and $h_{i,v}$ is the aggregated message passing through hyperedge $e$ at iteration $i + 1$.

It is not hard to see that B.6 can be realized by $B$ iterations of RelNN layers with Expand operations where $B$ is the max arity of hyperedges. This can be done by expanding each node feature to every high arity features that contain the node, and aggregate them at the tuple corresponding to each hyperedge. Then, B.7 can also be realized by $B$ iterations of RelNN layers with Reduce operations, as the tuple feature will finally be reduced to a single node contained in the tuple.

This approach has lower complexity compared to the GNNs we study applied on hyperedges, because it only requires communication between nodes and hyperedges connecting to them, which takes $O(|V| \cdot |E|)$ time at each iteration. Compared to them, RelNNs takes $O(|V|^B)$ time because RelNNs keep features of every tuple with max arity $B$, and allow communication from tuples to tuples instead of between tuples and single nodes. An example is provided below that this approach can not solve while RelNNs can.

Consider a graph with 6 nodes and 6 edges forming two triangles $(1, 2, 3)$ and $(4, 5, 6)$. Because of the symmetry, the representation of each node should be identical throughout hypergraph message passing rounds. Hence, it is impossible for these models to conclude that $(1, 2, 3)$ is a triangle but $(4, 2, 3)$ is not, based only on the node representations, because they are identical. In contrast, RelNNs with max arity 3 can solve them (as standard triangle detection problem in Table 1).

### B.4 PROOF OF THEOREM 3.1: ARITY HIERARCHY.

Morris et al. (2019) have connected high-dimensional GNNs with high-dimensional WL tests. Specifically, they showed that the GNNs on $B$-ary hypergraphs are equally expressive as $B$-dimensional WL test on graph isomorphism test problem. In Theorem B.5 we proved that GNNs on $B$-ary hypergraphs are equivalent to RelNN of maximum arity $B + 1$ in terms of expressiveness. Hence, RelNN of maximum arity $B + 1$ can distinguish if two non-isomorphic graphs if and only if $B$-dimensional WL test can distinguish them.

However, Cai et al. (1992) provided an construction that can generate a pair of non-isomorphic graphs for every $B$, which can not be distinguished by $(B-1)$-dimensional WL test but can be distinguished by $B$-dimensional WL test. Let $G_B^1$ and $G_B^2$ be such a pair of graph.

Since RelNN of maximum arity $B+1$ is equally expressive as GNNs on $B$-ary hypergraphs, there must be such a RelNN that classify $G_B^1$ and $G_B^2$ into different label. However, such RelNN can not be realized by any RelNN of maximum arity $B$ because they are proven to have identical outputs on $G_B^1$ and $G_B^2$.

In the other direction, RelNNs of maximum arity $B+1$ can directly realize RelNNs of maximum arity $B$, which completes the proof.

### B.5 PROOF OF THEOREM 3.4: UPPER DEPTH BOUND FOR UNBOUNDED-PRECISION RELNN.

The idea for proving an upper bound on depth is to connect RelNNs to WL-test, and use the $O(n^B)$ upper bound on number of iterations for $B$-dimensional test (Kiefer & Schweitzer, 2016), and FOC formula is the key connection.

For any fixed $n$, $B$-dimensional WL test divide all graphs of size $n$, $\mathcal{G}_{=n}$, into a set of equivalence classes $\{\mathcal{C}_1, \mathcal{C}_2, \cdots, \mathcal{C}_m\}$, where two graphs belong to the same class if they can not be distinguished by the WL test. We have shown that RelNNs of maximum arity $(B+1)$ must have the same input for all graphs in the same equivalence class. Thus, any RelNN of maximum arity $B+1$ can be view as a labeling over $\mathcal{C}_1, \cdots, \mathcal{C}_m$.

Stated by Cai et al. (1992), $B$-dimensional WL test are as powerful as $\text{FOC}_{B+1}$ in differentiating graphs graphs. Combined with the $O(n^B)$ upper bound of WL test iterations, for each $\mathcal{C}_i$, there must be an $\text{FOC}_{B+1}$ formula of quantifier depth $O(n^B)$ that exactly recognize $\mathcal{C}_i$ over $\mathcal{G}_{=n}$.

Finally, with unbounded precision, for any $f(n)$, RelNN of maximum arity $B+1$ and depth $f(n)$ can compute all $\text{FOC}_{B+1}$ formulas with quantifier depth $f(n)$. Note that there are finite number of such formula because the subscript of counting quantifiers is bounded by $n$.

For any graph in some class $\mathcal{C}_i$, the class can be determined by evaluating these FOC formulas, and then the label is determined. Therefore, any RelNN of maximum arity $B+1$ can be realized by a RelNN of maximum arity $B+1$ and depth $O(n^B)$.

### B.6 THE TIME AND SPACE COMPLEXITY OF RELNNS

Handling high-arity features and using deeper models usually increase the computational cost in terms of time and space. As an instance that use the architecture of RelNN, NLMs with depth $D$ and max arity $B$ takes $O(Dn^B)$ time when applying to graphs with size $n$. This is because both Expand and Reduce operation have linear time complexity with respect to the input size (which is $O(n^B)$ at each iteration). If we need to record the computational history (which is typically the case when training the network using back propagation), the space complexity is the same as the time complexity.

GNNs applied to $(B-1)$-ary hyperedges and depth $D$ are equally expressive as RelNNs with depth $O(D)$ and max arity $B$. Though up to $(B-1)$-ary features are kept in their architecture, the broadcast message passing scheme scale up the complexity by a factor of $O(n)$, so they also have time and space complexity $O(Dn^B)$. Here the length of feature tensors $W$ is treated as a constant.

### B.7 PROOF OF THEOREM 4.2: PAC BOUNDS FOR RELNNS

We first formally define the $(\epsilon, \delta)$ approximation on functions mapping real number factors.

**Definition B.3.** We say $M_1$ is an $(\epsilon, \delta)$-approximation of $M_2 : \mathbb{R}^W \to \mathbb{R}^W$ if and only if with probability at least $(1-\delta)$, $||M_1(x) - M_2(x)|| < \epsilon$.

The proof builds on bounding the error between the representations computed by $M$ and $M'$. Assuming the Lipschitz continuity with constant $\lambda$, we will show four facts that are sufficient to bound the error between $M$ and $M'$, and thus claim the approximation of $M$ on $f$.

- (a) Feed-forward networks increase the error by at most a factor of $\lambda$.
- (b) The min-max aggregation increase the error by at most a factor $\sqrt{W}$.
- (c) The mean aggregation keeps the error bounded.

- (d) The probability to avoid any failures is lower bounded by $1 - O(Dn^B\delta)$.

Fact (a) is guaranteed under the definition of Lipschitz continuity. For fact (b), we denote $t$ be the error bound on the representations before aggregation, and $t$ also bounds the error at each position. In the worst case, min-max aggregation will get the error $t$ at all positions, which implies a new error bound $\sqrt{W}t$ on the aggregated representation on the 2-norm. Fact (c) is guaranteed by the convexity of 2-norm. Fact (d) can be derived by a simple union bound on the failure probability on all internal representations (they are $O(Dn^B)$ representations to compute when applying the layer for $D$ iterations).

Note that at each iteration we will run all feed-forward network in parallel for all representations, and aggregate some $n$-tuple of representations. These operations increase the error bound by a factor of at most $\lambda\sqrt{W}$ for min-max aggregation and $\lambda$ for mean aggregation.

**Sample complexity for RelNNs with gradient decent.**    Arora et al. (2019) have derived a more fine-grained analysis on the of two-layer neural networks optimized by gradient decent, which is later extended by Xu et al. (2020) under the *sequential learning* setting i.e. we use a correct network instance $M'$ to generate the ground-truth outputs to supervise all feed-forward networks at each layer.

**Theorem B.6** (Xu et al. (2020), sample complexity for over-parameterized MLP modules)**.** Let $\mathcal{M}$ be an over-parameterized and random initialized two-layer MLP trained with gradient decent for a sufficient number of iterations. Suppose $g : \mathbb{R}^W \to \mathbb{R}^W$ with components $g(x)^{(i)} = \sum_j \alpha_j^{(i)} (\beta_j^{(i)\top} x)^{p_j^{(i)}}$, where $\beta_j^{(i)} \in \mathbb{R}^W$, $\alpha \in \mathbb{R}$ and $p_j^{(i)} = 1$ or $p_j^{(i)} = 2l$ ($l \in \mathbb{N}_+$). The sample complexity $\mathcal{C}_\mathcal{M}(g, \epsilon, \delta)$ is

$$\mathcal{C}_\mathcal{M}(g, \epsilon, \delta) = O\left(\frac{\max_i \sum_j p_j^{(i)} |\alpha_j^{(i)}| \cdot ||\beta_j^{(i)}||_2^{p_j^{(i)}} + \log(W/\delta)}{(\epsilon/W)^2}\right) \tag{B.8}$$

We combine Theorem B.6 with our theorem 4.2 to derive the sample complexity in order to get $(\epsilon, \delta)$-approximation. We can compute $\epsilon_{NN} = O(\frac{\epsilon}{\lambda^D W^{D/2}})$ and $\delta_{NN} = O(\frac{\delta}{Dn^B})$ for feed-forward network in the RelNN, and bring them as to Theorem B.6 to get the following equation

$$\mathcal{C}_\mathcal{M}(g, \epsilon, \delta, n) = O\left(\frac{\max_i \sum_j p_j^{(i)} |\alpha_j^{(i)}| \cdot ||\beta_j^{(i)}||_2^{p_j^{(i)}} + B\log(nDW/\delta)}{(\frac{\epsilon}{\lambda^D W^{D/2}})^2}\right) \tag{B.9}$$

where $g : \mathbb{R}^W \to \mathbb{R}^W$ has components $g(x)^{(i)} = \sum_j \alpha_j^{(i)} (\beta_j^{(i)\top} x)^{p_j^{(i)}}$, and $B, D, W$ are the dimension, number of iterations, width for specifying the model family $\mathcal{M}$.

The sample complexity does not have a requirement of the sample graph size, because under the *sequential learning* setting the feed-forward networks are trained under supervision on each node and tuple at each layer. Though this setting can not be implemented in practice, in our experiments we observe good generalization of models trained on small graphs.

## C    EXPERIMENTS

### C.1    EXPERIMENT SETUP

For all problems, we have 800 training samples, 100 validation samples, and 300 test samples for each different $n$ we are testing the models on.

We then provide the details on how we synthesize the data. For most of the problems, we generate the graph by randomly selecting from all potential edges i.e. the Erdős–Rényi model. We sample the number of edges around $n, 2n, n\log n$ and $n^2/2$. For all problems, with $50\%$ probability the graph will first be divided into $2, 3, 4$ or $5$ parts with equal number of components, where we use the first generated component to fill the edges for rest of the components. Some random edges are added afterwards. This make the data contain more isomorphic sub-graphs, which we found challenging empirically.

**Substructure Detection.** To generate a graph that does not contain a certain substructure, we randomly add edges when reaching a maximal graph not containing the substructure or reaching the edge limit. For generating a graph that does contain the certain substructure, we first generate one that does not contain, and then randomly replace present edges with missing edges until we detect the substructure in the graph. This aim to change the label from "No" to "Yes" while minimizing the change to the overall graph properties, and we found that data generated using edge replacing is much more difficult for neural networks compared to random generated graphs from scratch.

**Family Tree.** We generate the family trees using the algorithm modified from Dong et al. (2019). We add people to the family one by one. When a person is added, with probability $p$ we will try to find a single woman and a single man, get them married and let the new children be their child, and otherwise the new person is introduced as a non-related person. Every new person is marked as single and set the gender with a coin flip.

We adjust $p$ based on the ratio of single population: $p = 0.7$ when more than $40\%$ of the population are single, and $p = 0.3$ when less than $20\%$ of the population are single, and $p = 0.5$ otherwise.

**Connectivity.** For connectivity problems, we use the similar generation method as the substructure detection. We sample the query pairs so that the labels are balanced.

### C.2 MODEL IMPLEMENTATION DETAILS

For all models, we use a hidden dimension $128$ except for 3-dimensional HD-GNN and 4-dimensional NLM where we use hidden dimension $64$.

All model have $4$ layers that each has its own parameters, except for connectivity where we use the recurrent models that apply the second layer $k$ times, where $k$ is sampled from integers in $[2 \log n, 3 \log n]$. The depths are proven to be sufficient for solving these problems (unless the model itself can not solve).

All models are trained for 100 epochs using adam optimizer with learning rate $3 \times 10^{-4}$ decaying at epoch 50 and 80.

We have varied the depth, the hidden dimension, and the activation function of different models. We select sufficient hidden dimension and depth for every model and problem (i.e., we stop when increasing depth or hidden dimension doesn't increase the accuracy). We tried linear, ReLU, and Sigmoid activation functions, and ReLU performed the best overall combinations of models and tasks.

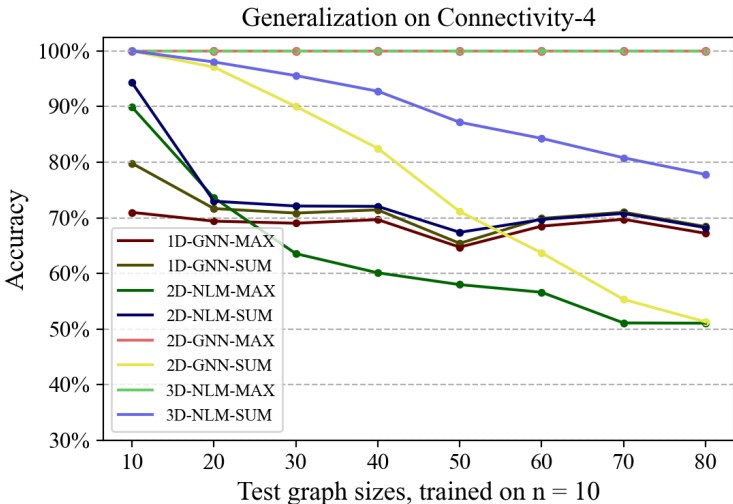

Figure 2: How the performance of models drop when generalizing to larger graphs on the problem connectivity-4 (trained on graphs with size 10).

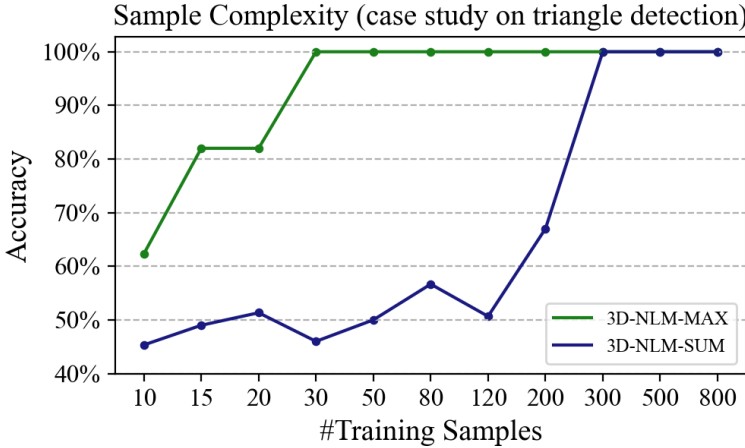

Figure 3: The test accuracy when the number of training samples varies between 10 and 300. All numbers are averaged over three runs with randomly selected training sets. Note that we do not test structural generalization here, so $n = 10$ for both training and testing).

### C.3 ADDITIONAL ABLATION STUDIES

**Case study on structure generalize.** We run a case study on the problem connectivity-4 about how the generalization performance changes when the test graph size gradually becomes larger. Figure 2 how how these models generalize to gradually larger graphs with size increasing from 10 to 80. From the curves we can see that only models with sufficient expressiveness can get 100% accuracy on the same size graphs, and among them the models using max aggregation generalize to larger graphs with no performance drop. 2-ary GNN and 3-ary NLM that use max aggregation have sufficient expressiveness and better generalization property. They achieve 100% accuracy on the original graph size and generalize perfectly to larger graphs.

**Case study on sample complexity.** In order to better visualize how using max and sum aggregation functions affect the sample complexity. We run an ablation study on the triangle detection problem where we try different numbers of training samples from 10 to 800. We test 3-ary NLM with max and sum aggregation which both reach 100% accuracy on the same graph size (we do not test generalization here since max aggregation is shown to be more generalizable than sum aggregation). Figure 3 indicates that the NLM model with max aggregation require much fewer training samples ($\sim 30$) to reach the perfect accuracy compare to sum aggregation ($\sim 300$.)

