# OpenReview forum: "On the Expressiveness and Learning of Relational Neural Networks on Hypergraphs"
_ICLR.cc/2022/Conference — ICLR 2022 Submitted_

### Official Review · Reviewer_d5Yc · 2021-10-27

**Correctness:** 3
**Technical Novelty And Significance:** 1
**Empirical Novelty And Significance:** 1
**Recommendation:** 3
**Confidence:** 5

**Main Review:**

The paper offers an interesting perspective on relational models in that it studies them via a unifying framework. Moreover, it generalizes existing results beyond the binary setting into arbitrary arity settings. However, the results presented in the paper are not significant enough. Indeed, the proposed arity hierarchy is a simple generalization of existing expressiveness results as well as results on the WL hierarchy, and such results are not surprising, as they are already established, e.g., for higher-order GNNs [1], which study the higher-arity setting. Moreover, the depth and arity requirements provided in Table 1 follow easily from the FOC^k characterization, and thus do not offer any new insights in my opinion. To illustrate, connectedness can clearly be captured with B=3, as the general connectedness task can be written as a formula in FOC^3 (even just FO^3). By contrast, FOC^2 is not sufficient to capture connectedness, and this propagates to RelNN with B=2 and fixed depth. Moreover, all S-T problems can be easily solved with a depth n GNN, simply by propagating a ``beacon'' from S to N and seeing whether this reaches T after n hops. The proposed conjectures are also not rigorous enough, and do not provide further insights as to the power of RelNN models.

I also find that the RelNN framework does not produce any novel understanding of relational models. Indeed, the connection between, e.g., GNNs and Transformers is already well-established and studied in the literature [2], and the current study does not propose any novelty here. Furthermore, the experimental analysis is not convincing, and merely confirms the aforementioned simple results, and does not study the learning bounds part of the paper with sufficient detail.

All in all, the paper makes a series of small contributions which, in my opinion, are not sufficient for publication.

[1] Nicolas Keriven and Gabriel Peyre. Universal Invariant and Equivariant Graph Neural Networks, NeurIPS 2019.

[2] William L. Hamilton. Graph Representation Learning. Morgan and Claypool Publishers, 2020.

**Summary Of The Paper:**

The paper proposes a unifying study of models for relational learning, namely Transformers, Graph Neural Networks (GNNs), and Neural Logic Machines (NLM), under the RelNN framework, and provides a series of results pertaining to their expressiveness relative to the arity of relations they support, as well as the depth, i.e., the number of iterations, they use during computation. In particular, arity and depth hierarchies are studied, and a set of arity and depth requirements are stated for a set of computational problems. Furthermore, the generalization ability of RelNN models is studied in the fixed-precision and infinite-precision settings. In the former setting, it is shown that there exists a finite size N for input graphs at which perfectly fitting all possible input graphs yields perfect generalization onto graphs of size greater than N. In the latter case, the sample complexity required to achieve certain error bounds is studied, and it is shown that more range-restricted aggregation function such as max and mean enable improved sample complexity. Finally, the paper conducts an experimental analysis on several synthetic problems, and achieves results validating the earlier provided arity and depth results for RelNN.

**Summary Of The Review:**

The paper's main contributions and results, especially on the expressiveness side, are simple and already well-studied in the literature, and thus are not insightful or novel enough to warrant publication.

---

> ### Author Response · Authors · 2021-11-22
> **Author Response to Reviewer d5Yc**
>
> Thank you for your helpful comments.
>
> **Q1:** Some results are already established in [1] and [2].
>
> **A1:** [1,2] studied the asymptotic lower and upper bounds of the max arity for universal graph invariant/equivariant functions. That is, if we want to build a GNN that can solve all graph problems with input graph size $n$, what arity do we need? Thus, the result is a function of $n$, specifically, $\frac{n-2}{2}$. Our work is fundamentally different from theirs. We study the problem hierarchy defined by constant max arity, and when we say a RelNN can solve a problem, we mean it can be applied to an arbitrarily large graph size $n$, which is a more practical setting.
>
> **Q2:** Table 1 follows easily from the FOC$^k$ characterization.
>
> **A2:** First, we would like to clarify that Table 1 is not our main result, but a list of example problems in our arity and depth hierarchy based on the best algorithms we know, serving as an intuition for "solving harder problems requires relational neural networks with more complex structures."
>
> **Arity characterization.** There are two significant differences between our work and [3]. First, they only proved $\mathrm{FOC}^2$ can be realized by GNNs. We have extended this proof to hyper-graphs: a (k-1)-ary hyper-graph GNN can realize $\mathrm{FOC}^k$. The proof is non-trivial, because message passing defined over hyper-graphs involves hyper-edge neighbors: message passing happens between two hyperedges that only differ at a node, such as $(v_1, \cdots, x, \cdots, v_k)$ and $(v_1, \cdots, y, \cdots, v_k)$. To do that, we first proved a bidirectional reduction between (k-1)-ary GNNs and our k-ary RelNN formulation. Then, it is more obvious to prove the equivalence between hyper-graph RelNNs and FOCs.
>
> Second, we proved a "necessary" condition for realizing $\mathrm{FOC}^k$. That is, (k-2)-ary GNN cannot realize $\mathrm{FOC}^k$. This is a helpful result because now we can determine whether a RelNN can solve a problem by considering whether a logic formula can represent the solution to this problem.
>
> **Depth characterization.** We don't know any existing work on the theoretical analysis of depth in relational neural networks. And we would like to kindly point out three points about the reviewer's argument.
>
> - For the specific S-T connectivity problem, it is obvious that it can be solved by a 2-ary GNN with $O(\log n)$ depth (RelNN[$\log n$,3]), but to our best knowledge, it is open whether a constant depth 1-ary or 2-ary graph neural network (e.g., 2) can solve it (i.e. whether it is in RelNN[2,2] or RelNN[2,3]). Our paper establishes a connection between GNN complexity and complexity classes in distributed computing (broadcasting networks).
> - In general, it is unknown whether a deeper GNN is more expressive than a shallower GNN. For example, it is unknown whether depth-3 GNN is more expressive than depth-2 ones; it is also unknown whether a depth-$\log n$ GNN is more expressive than a depth-2 GNN.
> - Moreover, there is an intriguing asymmetry between arity and depth: we can show that as the max arity increases, we get more and more powerful GNNs. However, when the arity is fixed to be $k$, increasing depth beyond $O(n^{k-1})$ will not further increase the power.
>
> **Q3:** The connection between GNNs and Transformers is already well established.
>
> **A3:** Thanks for pointing to the reference. The connection between GNNs and Transformers is an example we used to illustrate the notion of "equivalent expressiveness." One of our main theoretical results, however, is the reduction between message-passing GNNs and RelNNs on hyper-graphs (Theorem B.5).
>
> [1] Nicolas Keriven and Gabriel Peyre. Universal Invariant and Equivariant Graph Neural Networks. In NeurIPS, 2019.
>
> [2] William L. Hamilton. Graph Representation Learning. Morgan and Claypool Publishers, 2020.
>
> [3] Pablo Barceló, Egor V Kostylev, Mikael Monet, Jorge P ́erez, Juan Reutter, and Juan Pablo Silva. Thelogical expressiveness of graph neural networks. In ICLR, 2020.

---

> > ### Comment · Reviewer_d5Yc · 2021-11-27
> > **Reviewer Response**
> >
> > Dear Authors,
> > I have read your response, and am not convinced by your arguments or about the significance of the contributions. I understand that the technicalities in producing the proofs could be non-trivial, but I must stress that the results themselves remain incremental and add little to our understanding of GNNs. In particular, the results you show all follow rather intuitively from properties of the k-WL kernel, e.g., the fact that it needs n^k-1 iterations to provably stop yielding more refinements, and the correspondence between k-GNN and k-WL. Hence, I will keep my current recommendation.

---

### Official Review · Reviewer_m5XE · 2021-10-28

**Correctness:** 2
**Technical Novelty And Significance:** 3
**Empirical Novelty And Significance:** 1
**Recommendation:** 5
**Confidence:** 4

**Main Review:**


### **Strengths**

1) The paper is well organised.
2) RelNNs unify three neural network architectures on relational data (GNNs, NLMs, Transformers).
3) The paper studies expressiveness and generalisability of RelNNs.


### **Weaknesses**

1) An important contribution claimed in the paper is that RelNNs can be applied to hypergraph reasoning tasks. However, all tasks in the experiments are graph-related (i.e., without hyperedges of size more than two).
2) The experiments seem to not compare with existing methods, e.g., [Barcelo et al., ICLR'20]. Adding such a comparison would help making an empirical argument for why the results are significant.
3) All datasets are synthetically generated and it is unclear how the results of the paper can be generalised to real-world relational data (e.g., real-world graph classification problems such as molecule property prediction).
4) The synthetic datasets are tiny: all graphs contain fewer than 100 nodes.
5) There is no discussion of important prior work and it is unclear if RelNNs can generalise neural networks defined on hypergraphs. Some publications are listed below:
    1) Hypergraph Neural Networks, AAAI'19,
    2) HyperGCN: A New Method for Training Graph Convolutional Networks on Hypergraphs, NeurIPS'19,
    3) Hypergraph Convolution and Hypergraph Attention, Pattern Recognition'21,
    4) Hyper-SAGNN: a self-attention based graph neural network for hypergraphs, ICLR'20,
    5) Be More with Less: Hypergraph Attention Networks for Inductive Text Classification, EMNLP'20,
    6) UniGNN: a Unified Framework for Graph and Hypergraph Neural Networks, IJCAI'21.

**Summary Of The Paper:**

Neural Networks for relational data have been an active topic of research interest in recent years. There is still a lack of full understanding of the expressivity, generalisability of these models. The contributions of the paper are:
1) Development of relational neural networks (RelNNs) for (hyper)graphs that unify graph neural networks (GNNs), neural logical machines (NLMs), and transformers,
2) Analysis of the expressivity of RelNNs in terms of (i) maximum hyperedge size and (ii) depth of RelNN, and
3) A study of the generalisability of RelNNs to unseen test data.

**Summary Of The Review:**

Overall the paper is well organised with good theory.
However, the main claims made in the paper need rigorous empirical evaluation.
Positioning with missing prior work and empirical evaluation on real-world hypergraph reasoning tasks would greatly improve the quality of the paper.

---

> ### Author Response · Authors · 2021-11-22
> **Author Response to Reviewer m5XE**
>
> Thank you for your helpful comments.
>
> **Q1:** Experiments are on graphs rather than hypergraphs.
>
> **A1:** First, we would like to clarify that our hierarchy of problems depends on the minimum arity and depth requirements of RelNNs that can solve them, but not on the arity of inputs. Many problems on 2-ary graphs (without hyperedges, e.g., all-pair connectivity, triangle detection) can not be solved without modeling higher-order relations.
>
> Thank you for suggesting additional experiments.  Per your request, we have extended the reasoning task for family-tree problems to hyper-graphs, where the family relation is represented by (father, mother, child) hyper-edges rather than edges between the child and each of its parents. 3-ary NLM still gets 100% accuracy when trained and tested on $n=20$. It also generalizes to $n=80$. We have included this experiment in the paper.
>
> ---
> **Q2:** Missing comparison with existing methods.
>
> **A2:** The GNN we use is equivalent to ACR-GNN in Barceló et al.[7], which had the best expressiveness and performance in modeling FOC$_2$ properties in their setting.
>
> ---
> **Q3:** Datasets are synthetically generated and tiny.
>
> **A3:** We would like to clarify our contribution as building a theoretical framework for unifying common graph models and analyzing expressiveness and learning. We are not proposing RelNNs as a new powerful model for graph reasoning. Hence, we didn't compare with SOTA on popular graph datasets.
>
> It would be interesting to extend the experiments to realistic datasets. We didn't do that primarily because of the memory complexity of hypergraph RelNNs. A depth $D$, arity $B$ RelNN has the complexity $O(Dn^B)$, which is practically infeasible for large real-world graphs. However, this does not affect the significance of our theorems, as we focus on theoretical aspects of hyper-graph reasoning complexity. We leave efficiency improvements to existing hyper-graph neural networks as future work.
>
> ---
> **Q4:** Discussion of important prior work.
>
> **A4:** Thanks for suggesting these references. We have included discussions of these hypergraph relational neural networks in the revised paper. Below we summarize our results.
>
> First, all models mentioned by the reviewer [1-6] can be characterized as specific implementations of hyper-graph RelNNs. Specifically, [1-3, 5-6] use message passing between regular nodes and special nodes corresponding to hyper-edges. Hyper-SAGNN[4] focuses on predicting the existence of hyperedges based on the node representations. This can be viewed as a specific hyper-edge classification problem on graphs with only node features and no edge features. The full proof of these relationships has been included in Appendix B.2.
>
> Second, all models mentioned by the reviewer are less expressive than RelNNs with the same arity (e.g., they are weaker than the hyper-graph neural network studied in our paper [8]). Intuitively, this is because RelNNs allow communication between arbitrary tuples of nodes rather than only communication between hyperedges and their nodes. We provide a concrete example here:
>
> > Consider a graph with $6$ nodes and $6$ edges forming two triangles $(1,2,3)$ and $(4,5,6)$. Because of the symmetry, the representation of each node should be identical throughout hypergraph message passing rounds. Hence, it is impossible for these models to conclude that $(1,2,3)$ is a triangle but $(4,2,3)$ is not, based only on the node representations, because they are identical.
>
> Third, the benefit of the hyper-graph message-passing scheme used by [1-6] is computational efficiency: they keep only the features of existing edges and nodes, but not all tuples. But this is achieved at the cost of sacrificing expressiveness.
>
> **References**
>
> [1] Yifan Feng, Haoxuan You, Zizhao Zhang, Rongrong Ji, and Yue Gao. Hypergraph neural networks. In AAAI, 2019.
>
> [2] Naganand Yadati, Madhav Nimishakavi, Prateek Yadav, Vikram Nitin, Anand Louis, and ParthaTalukdar. Hypergcn: A new method of training graph convolutional networks on hypergraphs. In NeurIPS, 2019.
>
> [3] Song Bai, Feihu Zhang, and Philip HS Torr.  Hypergraph convolution and hypergraph attention. Pattern Recognition, 110:107637, 2021.
>
> [4] Ruochi Zhang, Yuesong Zou, Jian Ma. Hyper-SAGNN: a self-attention based graph neural network for hypergraphs. In ICLR, 2020.
>
> [5] Kaize Ding, Jianling Wang, Jundong Li, Dingcheng Li, and Huan Liu. Be more with less: Hypergraphattention networks for inductive text classification. In EMNLP, 2020.
>
> [6] Jing Huang and Jie Yang. Unignn: a unified framework for graph and hypergraph neural networks. In IJCAI, 2021.
>
> [7] Pablo Barceló, Egor V Kostylev, Mikael Monet, Jorge P ́erez, Juan Reutter, and Juan Pablo Silva. Thelogical expressiveness of graph neural networks. InICLR, 2020.
>
> [8] Christopher Morris, Martin Ritzert, Matthias Fey, William L Hamilton, Jan Eric Lenssen, GauravRattan, and Martin Grohe. Weisfeiler and Leman go neural: Higher-order graph neural networks. In AAAI, 2019.

---

### Official Review · Reviewer_NLAa · 2021-11-02

**Correctness:** 4
**Technical Novelty And Significance:** 2
**Empirical Novelty And Significance:** 2
**Recommendation:** 5
**Confidence:** 2

**Main Review:**

On the positive side: 1) the authors introduce quite a general class of trainable models, and 2) they characterise the expressiveness and the generalisation capacity of these models. Noteworthy is the notion of quantifying the generalisation as a function of the size of the input test graphs.
On the negative side: the theoretical analysis is not accompanied by an adequate empirical investigation.
First of all, it is not clear if the graphs in the experiments are labelled or unlabelled: if they are labelled it would be of interest to report the label alphabet size and distribution; if the results refer to unlabelled graphs, it should be of interest to extend the experimentation to the more realistic case of labelled graphs.
Since one of the main contribution is the characterisation of the sample complexity and the generalisation to larger graphs, it would be appropriate to report an extensive experimental analysis of the alignment between theory and practice; it would therefore be of interest to report: 1) learning curves (i.e. reporting on the horizontal axis the training set size) to analyse the sample complexity behaviour in various scenarios  and 2) an analysis of the performance as the test set graphs size increases.
Presenting result on some more challenging tasks would also be of interest, e.g. a graph diameter regression task.
Please consider using an adequate testing of the significance of the result differences (e.g. following [Benavoli, Alessio, Giorgio Corani, Janez Demšar, and Marco Zaffalon. "Time for a change: a tutorial for comparing multiple classifiers through Bayesian analysis." The Journal of Machine Learning Research 18, no. 1 (2017): 2653-2688.] or using the library [https://github.com/sherbold/autorank]).


**Summary Of The Paper:**

The authors present a framework to study the expressiveness and the generalisation capacity of a class of forward message passing algorithms that encompass graph neural networks, transformers and neural logic machines.

**Summary Of The Review:**

The theoretical analysis is not accompanied by an adequate empirical investigation.

---

> ### Author Response · Authors · 2021-11-22
> **Author Response to Reviewer NLAa**
>
> Thank you for your helpful comments.
>
> **Q1:** Labeled and unlabeled graphs.
>
> **A1:** We assume that by "labeled graphs," you meant different nodes and edges have different input representations. We indeed used labeled graphs: in all of our family tree problems, nodes have labels indicating their genders, and edges have labels indicating their relationships (such as mother-child or father-child).
>
> It would be interesting to extend the experiments to realistic datasets. We didn't do that primarily because of the memory complexity of hypergraph RelNNs. A depth $D$, arity $B$ RelNN has the complexity $O(Dn^B)$, which is practically infeasible for large real-world graphs. However, this does not affect the significance of our theorems, as we focus on theoretical aspects of hyper-graph reasoning complexity. We leave efficiency improvements to existing hyper-graph neural networks as future work.
>
> ---
>
> **Q2:** Experiments.
>
> **A2:** Thanks for your suggestions. We have added the [learning curves](https://drive.google.com/uc?export=download&id=1o5NNNJeHRLqLkWGCcQ0s956dW6mXsqkS) for sample complexity and the [generalization performance](https://drive.google.com/uc?export=download&id=143xEjWPTgLx1pQsfFXdgIpl_BGoJpajv) to varying sizes of graphs.
>
> As for more challenging tasks and more adequate comparisons between classifiers, please refer to our answer A1.

---

### Official Review · Reviewer_6v52 · 2021-11-03

**Correctness:** 2
**Technical Novelty And Significance:** 3
**Empirical Novelty And Significance:** 2
**Recommendation:** 3
**Confidence:** 3

**Main Review:**

Pros:
1.	The proposed hierarchical architecture of RelNN presents a novel angel to understand relational neural networks.
2.	Its proof on RelNN’s improved expressiveness with larger D and B seems correct, and it brings meaningfully new knowledge.

Cons:
1.	Questionable claim. RelNN’s claimed unification of GNN, Transformers, NLM seems far-fetched and lacks justification. It is true that most GNNs can be conceptually viewed as a message passing process involving successive updates of edge features and then of node features. While I have no problem with the resemblance, how GNN’s computation graph can be rigorously expressed using RelNN’s framework is completely unjustified. It would be very helpful to explicitly explain what each T_{i,j} in Fig. 1 refers to for at least two concrete instances of GNN (GCN and GAT for example), as is also the case with Transformer. The correctness of this claim on unification appears questionable to me before more details are confirmed.
2.	Limited Readability. Besides the first point, there are many other places where the reasoning seems ambiguous or even arbitrary with abused terms and typos. For example in Sec.2.1, it first defines hypergraph representation functions as X = {X_0, X_1, X_2 …} and then in the example right below it uses X = {X_2}, which contradicts; in the same subsection it mentions “hyperedge representation function Y on V”, which should be “hypergraph representation function Y on the set of all size-k tuples of V’s nodes”; In the third bullet point,  Y = {X_1} should be Y = {Y_1}. In sec. 2.2, the "j+1 dimension" should be (j+1)st and it should be clarified that the feature dimension is the last dimension of the tensor. There are many other such places throughout the paper. While some are easy to sort out the actual meaning based on context, some others take readers significant time to figure out or even to realize. Therefore, the general readability is very limited.
3.	Experiments and Scalability. The paper claims main contribution on hypergraphs, but the experiments are limited to graph datasets only. Meanwhile, only n=80 is tested. Can RelNN work when n is large? It seems that RelNN struggles seriously with slightly larger graphs or hypergraphs with large hyperedges because of its node tuple definition. A thorough analysis here is very necessary. It also appears unclear what a 3-ary GNN is. Is that a realization of RelNN with B = 3? That also comes down to my first point what GNN is instantiated using RelNN.


**Summary Of The Paper:**

This paper presents a new generic framework of relational neural network defined over hypergraphs, which they call RelNN. RelNNs take an architecture which hierarchically operates on representation functions of node tuples of different sizes and can be stacked into multiple layers. The framework is claimed to have unified graph neural network, neural logical machines and tranformers. The authors conduct analysis of RelNNs and show that its expressiveness increases as the artity or the number of RelNN layers grows sufficiently large. The paper also characterizes the generalizability of RelNN under certain conditions.

**Summary Of The Review:**

In summary, I appreciate that the authors make a great effort to unify several important relational neural networks. However, the paper suffers from its weak justification (on the claimed unification), lack of scalability, insufficient experiment, and ambiguous writing style in general. Therefore, I do not vote for its acceptance.

---

> ### Author Response · Authors · 2021-11-22
> **Author Response to Reviewer 6v52**
>
> Thank you for your helpful comments.
>
> **Q1:** How is GNN's computation graph expressed by RelNN.
>
> **A1:** Xu et al. [1] have shown that fully connected GNNs using sum aggregation (called GIN in their paper) are provably the most expressive model among GNNs of the same arity, so common GNN variants including GCNs and GATs can be expressed by GINs.
>
> In the main paper, we provided an intuitive example that 2-ary GNNs are in the family of 3-ary RelNNs. The proof for the higher-arity case is non-trivial and is in Appendix B.2 (Theorem B.5.) The reduction is bidirectional: using expand and reduce operations, we can implement the hypergraph message-passing in GNNs.  Meanwhile, $k$-ary GNNs can realize all possible functions that can be computed by  $(k+1)$-ary RelNNs, and they have the same computation complexity.
>
> **Q2:** Readability.
> **A2:** Thanks for your comments; we have adjusted our notation based on your suggestions. As a specific example, when we said $X = {X_2}$, we meant that the input representation only contains binary relationships between nodes, but no nullary, unary, ternary, ... ones.  We hope we have clarified our exposition.
>
> ---
>
> **Q3:** Experiments on graphs but not hyper-graphs.
>
> **A3:** First, we would like to clarify that our hierarchy of problems depends on the minimum arity and depth requirements of RelNNs that can solve them, but not on the arity of inputs. Many problems on (2-ary) graphs (e.g., connectivity, bipartiteness) cannot be solved without modeling higher-order relations.
>
> Thank you for suggesting additional experiments.  Per your request, we have extended the reasoning task for family-tree problems to hyper-graphs, where the family relation is represented by (father, mother, child) hyper-edges rather than edges between the child and each of its parents. 3-ary NLM still gets 100% accuracy when trained and tested on $n=20$. It also generalizes to $n=80$. We have included this experiment in the paper.
>
> ---
>
> **Q4:** Scalability.
>
> **A4:** The complexity for running RelNNs with max arity $B$ and depth $D$ is $O(Dn^B)$ where $n$ is the size of the graph, so running high-arity RelNN models on a large graph can be very expensive. In the future, this can be potentially resolved by using "sparse" GNNs that do not compute for every high-order tuples, but these extensions are out of the scope of this paper.
>
> To clarify, 3-ary GNNs are GNNs applied to 3-ary hyperedges, which can be expressed by a RelNN with max arity 4. We will add these explanations in our revision.
>
> [1] Keyulu Xu, Weihua Hu, Jure Leskovec, and Stefanie Jegelka.   How powerful are graph neural networks?  In  ICLR, 2019.

---

### Official Review · Reviewer_4AVA · 2021-11-03

**Correctness:** 3
**Technical Novelty And Significance:** 2
**Empirical Novelty And Significance:** 3
**Recommendation:** 5
**Confidence:** 4

**Main Review:**

Strengths:
The studied problem of improving the representation and model generalization ability is important for many relational learning tasks.

The theoretical discussion of the unified relational learning framework is provided with theoretical analysis results.

Weaknesses:

State-of-the-art relational learning models are missing in the performance comparison, which can hardly demonstrate the effectiveness of the new learning framework. For example, the new framework should be compared against state-of-the-art graph neural models for capturing graph structural relationships, and the Transformer-based sequence learning.

The model scalability should be investigated in the experiments. Since the new model is built over the hypergraph structure, how is the time complexity of the hypergraph-guided relational learning should be investigated.

Lack of detailed description with respect to hyperparameter tuning strategies on the new proposed framework. Different hyperparameter selection may provide different prediction performance, which has been demonstrated in previous research work on graph neural networks and Transformer. It is necessary to present how the models are tuned to achieve good performance under a fair experimental setting.

**Summary Of The Paper:**

This paper explores the expressiveness of neural network-based relational learning models, such as graph neural networks and Transformer. To this end, a rational neural network is built upon the hypergraph structures to unify different models. Synthetic datasets are used for performance evaluation.

**Summary Of The Review:**

Several concerns for this work: (1) Key experimental setting information is missing in the evaluation section. (2) Many state-of-the-art relational learning methods are missing. (3) For fair evaluation, it is better to present the hyperparameter tuning for the new method.

---

> ### Author Response · Authors · 2021-11-22
> **Author Response to Reviewer 4AVA**
>
> Thank you for your helpful comments.
>
> **Q1:** SOTA relational learning models.
>
> **A1:** Our contribution is not a concrete model or a new learning framework, but rather, a theoretical analysis of relational neural network architectures and learning. Thus, we have been specifically focusing on representative models, although different concrete architecture designs (e.g., using attention) may change the sample efficiency of algorithms, they will not change the expressiveness. For example, we have proved that there do not exist graph neural networks that do not consider higher-order relationships (3-ary and 4-ary) that can solve the 4-Clique prediction task.
>
> ---
>
> **Q2:** Scalability.
>
> **A2:** The time complexity of computing a Neural Logical Machine with depth $D$ and max arity $B$ is $O(Dn^B)$ where $n$ is the number of nodes in the graph, which is the same as for graph neural networks applied to $(B-1)$-ary hypergraphs with depth $D$ (they are both RelNNs with max arity $B$). In general, higher arity and larger depth require more computational resources. We have added a discussion of this point in the revision.
>
> ---
>
> **Q3:** How are the hyperparameters tuned.
>
> **A3:** We have varied the depth, the hidden dimension, and the activation function of different models. We select sufficient hidden dimension and depth for every model and problem (i.e., we stop when increasing depth or hidden dimension doesn't increase the accuracy). We tried linear, ReLU, and Sigmoid activation functions, and ReLU performed the best overall combinations of models and tasks. We have included these details in the appendix in our revision.

---

### Author Response · Authors · 2021-11-22
**General Response to Reviewers**

We thank all reviewers for their insightful reviews and helpful, constructive comments. In our general response, we would like to clarify and reiterate the motivation and the contribution of this paper.

Suppose we have a relational reasoning problem to solve, for example, determining whether a graph is bipartite, an important property of a graph.
What structural hyperparameters of a relational neural network should I use? Can I solve it with a binary graph neural network (e.g., graph-isomorphism network (GIN) [1]) with 10 layers? How should I train it? After training, will it generalize to larger graphs?

This paper tries to answer these questions.

We have seen great success of various relational neural networks applied to reasoning over graph-structured data, studies on relationships between certain types of relational neural networks (e.g., GNNs and Transformers), and problems they can solve (a certain type of GNN can realize FOC$_2$ [2]). However, to the best of our knowledge, answers to the following theoretical questions about the expressiveness and learning capacity of hyper-graph GNNs for relational reasoning are missing from related literature.

- Expressiveness: Typically, people use graph neural networks only with binary edges and a finite number of layers. What's the "upper bound" for such networks? What are the problems that they cannot solve, even with a sufficiently large hidden dimension?
- Expressiveness: Can we improve the expressiveness of a relational neural network by increasing the arity of the intermediate features (even if the inputs only contain binary relationships)? If so, what are the limits of this?
- Expressiveness: Can we improve the expressiveness of a relational neural network by increasing the number of layers, especially specifying the number of layers as a function of the graph size?  If so, what are the limits of this?
- Learning: Let's say we have chosen correct structural hyperparameters (arity, depth) so that the desired solution can be represented by our relational neural network. But if we want to train it using a finite set of data, will it generalize to larger graphs that it has never seen?
- Learning: Let's say we have chosen correct structural hyperparameters (arity, depth) so that the desired solution can be represented by our relational neural network. But if we want to train it using a finite set of data, how much data do we need?

These questions involve studying relational neural networks with hyper-edges, and with non-constant depth (depending on the graph size). These questions are important (they allow us to build more powerful models and train them better) but not studied in the existing literature.

---

> ### Author Response · Authors · 2021-11-22
> **General Response to Reviewers - cont.**
>
> Concretely, we make the following contributions. The first group of contributions is about expressiveness.
>
> - Instead of proposing a novel model, we aim at developing a framework that can unify existing models (Hyper-graph GNNs, NLMs, Transformers). We developed formal reductions between them, which is non-trivial in the hyper-graph cases. This ensures that all our answers to the aforementioned questions apply to all models.
> - We formally proved the "if and only if" conditions for the expressive power w.r.t. the arity. That is, k-ary hyper-graph relational neural networks are sufficient and necessary for realizing FOC-k (formally, k-ary RelNN can realize FOC-k, but not (k-1)-ary RelNN). This is a helpful result because now we can determine whether a RelNN can solve a problem by trying to understand whether a logic formula can represent the solution to this problem.
> - We formally described the relationship between expressiveness and non-constant-depth RelNNs. We proposed a conjecture about the "depth hierarchy," and we connect the potential proof of this conjecture to distributed computing literature. Note that this is a very hard thing to prove: to the best of our knowledge, there is no proof for whether "a depth-2 3-ary relational neural network can solve S-T connectivity," although, superficially, it seems impossible.
> - We also developed an upper bound for RelNNs. This is a very interesting result: applying a k-ary hyper-graph RelNN for more than $O(n^{k-1})$ iterations has no effect on the expressiveness.
>
> The second group of contributions is about learning.
>
> - We prove, under certain realistic assumptions, it is possible to train a RelNN on a finite set of graphs, and it will generalize to arbitrarily large graphs. This is an outcome due to the weight-sharing nature of RelNNs.
> - We proved a finer-grained sample-complexity bound for RelNNs. This result theoretically suggests that if our goal is to realize a first-order logic formula, using max as the aggregation function is provably better than using sum as the aggregation function.
>
> Finally, we want to make a few short clarifications about specific confusions of the reviewers.
>
> **RelNN is not proposed as a practical neural network for improving performance on graph reasoning problems.** The experiments are not designed for comparing performance, but rather, are for empirical demonstration of the theorems and for exploring the tightness of the sample-complexity bounds and effectiveness of different aggregation functions.
>
> **Insights on graph model design.** We hope our work can serve as a foundation for designing relational neural networks: to solve a specific problem, what arity do you need? What depth do you need? What aggregation function should you use? Will my model have structural generalization (i.e., to larger graphs)?
>
> Consider the case where you want to classify whether a graph is bipartite. Our expressiveness theorem shows that you need at least 3-ary edges within the computational process (even if in this case the inputs are all binary) and thus a graph-isomorphism network will not do, and that you need $O(\log n)$ iterations but not a finite depth (e.g., 10) (which is a conjectured lower bound in theoretical distributed computing and is still an open problem). Our learning theorem shows that if you choose the correct architecture (3-ary, $\log n$ depth), your model will provably generalize to larger graphs. And, using max as the aggregation function has better sample complexity in this case.
>
> Of course, there are many other questions that remain unanswered, especially about the learning algorithm: what hidden dimension should I choose? What optimizer should I use? Some of these questions are illustrated with empirical experiments.
>
> [1] Keyulu Xu, Weihua Hu, Jure Leskovec, and Stefanie Jegelka.  How powerful are graph neural networks? In ICLR, 2019
>
> [2] Pablo Barceló, Egor V Kostylev, Mikael Monet, Jorge P ́erez, Juan Reutter, and Juan Pablo Silva. The logical expressiveness of graph neural networks. In ICLR, 2020

---

### Decision · Program_Chairs · 2022-01-20

**Decision:**

Reject

**Comment:**

The paper aims to improve our understanding of GNNs for relational reasoning. In this regard, authors develop a conceptual framework unifying popular models (GNNs, Transformers, etc.) for analyzing their expressiveness and learning capacity. We thank the reviewers and authors for engaging in an active discussion. Based on author comments, the goal of the paper was more of a conceptual exposition, however this did not come across to the reviewers from the manuscript at first. Thus, a better presentation would definitely make the paper much more accessible and useful to the community. Moreover, there were some concerns about the significance of the exposition and better positioning would help (e.g., how the results help improve our understanding of GNNs). Thus, unfortunately I cannot recommend an acceptance of the paper in its current form.